

# Statistical characteristics of raindrop size distribution over Western Ghats of India: wet versus dry spells of Indian Summer Monsoon

*Uriya Veerendra Murali Krishna[1], Subrata Kumar Das[1*], Ezhilarasi Govindaraj Sulochana[2], Bhowmik Utsav[1], Sachin Madhukar Deshpande[1], and Govindan Pandithurai[1]*

[1]Indian Institute of Tropical Meteorology, Pashan, Pune-411008, India

[2]College of Engineering, Guindy, Chennai-600 025, India

[*]*Correspondence to:* Subrata Kumar Das (skd_ncu@yahoo.com).

## Abstract:

The nature of raindrop size distribution (DSD) is analyzed during wet and dry spells of the Indian Summer Monsoon (ISM) over Western Ghats (WGs) using Joss-Waldvogel Disdrometer (JWD) measurements. The observed DSDs are fitted with gamma distribution, and the characteristic DSDs are studied during the summer monsoon seasons (June-September) of 2012-2015. The DSD spectra show distinct diurnal variation during wet and dry spells. The dry spells exhibit a strong diurnal cycle with two peaks, while the diurnal cycle is not prominent in the wet spells. The observational results reveal the microphysical characteristics of warm rain during both the wet and dry spells. Even though the warm rain processes are dominant over WGs during monsoon, the underlying dynamical processes cause the differences in DSD characteristics during wet and dry spells. In addition, the differences in DSD spectra with different rain rates are also observed during the wet and dry spells. The DSD spectra are further analyzed by separating into stratiform and convective types. Finally, an empirical relation between slope parameter, $\Lambda$ and shape parameter, $\mu$ is derived by best fitting the quadratic polynomial



for the observed data during both wet and dry spells as well as for the stratiform and convective types of precipitation. The $\Lambda$-$\mu$ relations obtained in the present study are slightly different in comparison with the earlier studies.

**Keywords:** Raindrop size distribution, Wet and dry spells, Monsoon, Western Ghats, Disdrometer.

## 1. Introduction

Western Ghats (WGs) is one of the two heavily rainfall regions in India. It receives a large
amount of rainfall (~6000mm) during the Indian Summer Monsoon (ISM; Das et al., 2017, and references therein). The monsoon rainfall in this region is contributed by both shallow clouds on the windward side (Kumar et al., 2013; Das et al., 2017; Utsav et al., 2017, 2019) and deep convection in the leeward side (Utsav et al., 2017, 2019; Maheskumar et al., 2014). In addition, thunderstorms also occur over WGs; however, they are very few during the monsoon period. The rainfall distribution over
WGs is complex in which topography plays a major role (Houze et al., 2012, and references therein). The distribution of rainfall over WGs depends on the region, whether it is on the windward side or leeward side of the mountains. These different properties correspond to different physical mechanisms. The intense rainfall in the windward side of the mountains, usually called the orographic precipitation comes from shallower clouds with long-lasting convection (Das et al., 2017; Utsav et al., 2019). One of
the major issues in precipitation measurements over WGs is the unavailability of a stable platform.

The ISM shows large spatial and temporal variability. It is well known that during the active (with good rainfall) and break (little or no rainfall) spells of the ISM, there are contrasting behaviors in

the formation of weather systems and large-scale instability. The strength of the ISM rainfall depends

on the frequency and duration of active and break spells (Kulkarni et al., 2011). This intra-seasonal

oscillation of precipitation is considered as one of the most important sources of weather variability

over the Indian region (Hoyos and Webster, 2007). From the earlier studies of Ramamurthy (1969),

active and break spells of the ISM have been extensively studied, especially during the last two decades

(Goswami and Ajaya Mohan, 2001; Gadgil and Joseph, 2003; Uma et al., 2011; Rajeevan et al., 2012;

Mohan and Rao, 2012; Das et al., 2013; Rao et al., 2016). The characteristic features of these active and

break spells have been well understood; for example, their identification (Rajeeven et al., 2006;

Rajeevan et al., 2010), spatial distribution (Ramamurthy, 1969; Rajeevan et al., 2010), circulation

patterns (Goswami and Ajaya Mohan 2001; Rajeevan et al., 2010), vertical wind and thermal structure

(Uma et al., 2011), rainfall variability (Deshpande and Goswami, 2014; Rao et al., 2016) and the macro-

and micro-physical features of clouds (Rajeevan et al., 2012; Das et al., 2013). Despite the fact that

different dynamical mechanisms for the observed rainfall distribution during wet and dry spells are well

understood, the microphysical processes of rain formation is still lacking.

Raindrop size distribution (DSD) is a fundamental microphysical property of the precipitation.

DSD is related to hydrometeor condensation, coalescence, and evaporation, which are important

parameters affecting the microphysical processes in the parameterization schemes of the numerical

weather prediction models (Gao et al., 2011). Hence, numerous observations of DSD during different

types of precipitation, different seasons, and different intra-seasonal periods at different locations are

essential for better representation of physical processes in the parameterization schemes. As a result, the

numerical weather prediction model communities are continuing their efforts to improve the simulation

of clouds and precipitation at the monsoon intra-seasonal scales by better representing the

microphysical processes through parameterization schemes. Different DSD characteristics lead to

different reflectivity ($Z$) and rainfall rate ($R$) relations. Hence, understanding the variability in DSD is

important to improve the reliability and accuracy in the quantitative precipitation estimation from radars

and satellites (Rajopadhyaya et al., 1998; Atlas et al., 1999; Viltard et al., 2000; Ryzhkov et al., 2005).

The active and break spells over WGs are nearly identical with the active and break phases over

the core monsoon zone (Gadgil and Joseph, 2003). The distribution of convective clouds over WGs

exhibit distinct spatiotemporal variability at intra-seasonal time scales (wet: analogous to active period

of ISM and dry: analogous to break period of ISM) during the ISM. Utsav et al. (2019) studied the

characteristics of convective clouds over WGs using X-band radar observations. Their study reveals that

the wet spells are associated with negative geopotential height anomalies at 500 hPa, negative OLR

anomalies and positive precipitable water anomalies. All these features promote the anomalous south-

westerlies, which favors the growth of convective elements over WGs. In contrast, a positive

geopotential height anomalies, positive OLR anomalies, and negative precipitable water anomalies are

observed during dry spells. This suppresses the convective activity over the Arabian Sea, and hence

little to no rain is observed over WGs during dry spells. These different dynamical properties affect the

convection during wet and dry spells over WGs.  However, the DSD (often used to speculate the

microphysical processes of rain) during wet and dry spells are least addressed, especially over WGs.

Climatological studies of DSD at several locations in a given region are rare, especially over

WGs. A few attempts have been made to understand the DSD characteristics over WGs. For example,

Konwar et al. (2014) studied the DSD characteristics by fitting three-parameter gamma function during

monsoon season. However, their study utilized only two season's data. Harikumar (2016) studied the

differences between DSD over coastal and high altitude station located in the WGs using the lognormal

fit to rain DSD. Das et al. (2017) studied the DSD characteristics during different precipitating systems

over WGs using Disdrometer and Micro Rain Radar measurements. They noticed different reflectivity

and rainfall rate relations during different types of precipitation. Sumesh et al. (2019) studied the DSD

differences between mid- and high-altitude regions in southern WGs during bright band events. These

studies were confined to DSD variations at different locations in WGs and/or during different types of

precipitation without the consideration of intra-seasonal variations. Hence, investigation of

microphysical characteristics of rain is still lacking especially during wet and dry spells over WGs.

The significance of different dynamical processes on the rainfall mechanisms during wet and dry

spells over WGs motivates to study the raindrop size distribution (which provides an indirect inference

on rain microphysical processes such as collision, coalescence, breakup, evaporation, etc. that can shape

the DSD) at intra-seasonal time scales. With this background, in the present study, we made an attempt

to address the following issues:

1. How does the DSD of orographic precipitation vary during wet and dry spells over WGs?

2. Does the wet and dry spell rainfall have different microphysical origin over the complex

terrain, WGs?

3. Does the DSD show any diurnal differences like rainfall distribution during wet and dry

spells over WGs?

4.  Establish the best fit for $\mu$-$\Lambda$ relationships during wet and dry spells.

To the knowledge of the authors, this is the first attempt to address the DSD variations during

wet and dry spells over the WGs region. The paper is organized as follows: the details of the instrument

and dataset used are presented in section 2. The methodology adopted for the separation of rainy days

into wet and dry spells is given in section 3. The observational results of DSDs during wet and dry

spells and the possible reasons are reported in section 4. The summary of this study is provided in

section 5.

## 2. Instrument and Datasets

Four year (2012- 2015) Joss-Waldvogel Disdrometer (JWD) measurements during the monsoon

months (June to September) at the High Altitude Cloud Physics Laboratory (HACPL), Mahabaleshwar

(17.92°N, 73.6°E, ~1.4 km above mean sea level) in the WGs is utilized to understand the DSD

variations during wet and dry spells of the ISM.

Joss-Waldvogel Disdrometer (Joss and Waldvogel, 1969) is an impact type disdrometer, which

measures the hydrometeors with a size ranging from 0.3 to 5.1 mm and arranges them in 20 channels

with 1-min integration time. JWD estimates the diameters of hydrometeors by sensing the voltage

induced by the downward displacement of 50 cm$^2$ styrofoam cone, once it is hit by the hydrometeors.

The accuracy of JWD measurements is 5%. The JWD has several shortcomings, such as noise,

sampling errors, and wind, etc. (Tokay et al., 2001; Tokay et al., 2003). JWD miscounts raindrops in the

lower size bins, specifically for drop diameters below 1 mm (Tokay et al., 2003). This is minimized

using the error correction matrix provided by the manufacturer. To reduce the sampling error arising

due to insufficient drop counts at lower rain rates, the rain rates less than 0.1 mm/h are discarded in the



present study. During heavy rain, JWD underestimates the number of smaller drops, known as disdrometer dead time. We didn't apply the dead time correction as it is not universally utilized (Tokay et al., 2001). Further, JWD cannot detect the raindrops with a diameter larger than 5.5 mm. However, for the present study, the raindrop diameters are mostly confined below 5.5 mm diameter over WG.

Hence, this may not affect the present analysis.

The concentration of raindrops, $N(D)$ (mm$^{-1}$ m-$^3$) at an instant of time is

$$N(D) = \Sigma_{i=1}^{20} \frac{n_i}{A\,\Delta t\,v(D_i)\,\Delta D_i} \qquad (1)$$

where $A$ is the surface area of observation, $t$ is the integration time, $n_i$ is the number of raindrops in the size class $i$, and $D_i$ is the mean diameter of size class $i$. $v(D_i)$ is the terminal velocity of the

raindrop in the $i$ channel and is estimated from Gunn and Kinzer (1949) as

$$v(D_i) = 9.65 - 10.3\,e^{-6\,D_i} \qquad (2)$$

JWD estimates rain rate ($R$) and reflectivity ($Z$) by assuming that the momentum is entirely due to the terminal fall velocity of the raindrops and the raindrops are spherical and expressed as

$$R = \frac{\pi}{6}\frac{3.6}{10^3}\frac{1}{A \times t}\sum_{i=1}^{20}(n_i\,D_i^3) \qquad (3)$$


$$Z = \Sigma_{i=1}^{20} N(D_i){D_i}^6\,\Delta D_i \qquad (4)$$

The one-minute DSD measurements obtained from JWD are fitted with a three-parameter gamma distribution, as suggested by Ulbrich (1983). The details about the DSDs used in the present study can be found in Das et al. (2017) and Krishna et al. (2017).

The functional form of the gamma distribution is





$$N(D) = N_\circ \, D^\mu \, e^{(-\Lambda D)} \tag{5}$$

Where, $N(D)$ is the number of drops per unit volume per unit size interval, $N_\circ$ ( in m$^{-3}$ mm$^{-(1+\mu)}$) is the number concentration parameter, $D$ (in mm) is the drop diameter, $\mu$ (unitless) is the shape parameter and $\Lambda$ (mm$^{-1}$) is the slope parameter of DSDs (Ulbrich, 1983; Ulbrich and Atlas, 1984). The gamma DSD parameters are calculated using moments proposed by Cao and Zhang (2009). Here, 2$^{nd}$, 3$^{rd}$ and 4$^{th}$

moments are utilized to estimate the Gamma parameters. This method gives relatively fewer errors compared to other methods (Konwar et al., 2014). The '$n$' order momentum of the distribution can be calculated as

$$M_n = \int_0^\infty D^n \, N(D) \, dD \tag{6}$$

The shape parameter, $\mu$, and the slope parameter, $\Lambda$ are given by

$$\mu = \frac{1}{(1-G)} - 4 \tag{7}$$

$$\Lambda = \frac{M_2}{M_3}(\mu + 3) \tag{8}$$

Where

$$G = \frac{M_3^2}{M_2 \, M_4} = \frac{\left[\int_0^\infty D^3 \, N(D) \, dD\right]^2}{\left[\int_0^\infty D^2 \, N(D) \, dD\right]\left[\int_0^\infty D^4 \, N(D) \, dD\right]} \tag{9}$$





The other parameters, normalized intercept parameter, $N_w$ (in mm$^{-1}$ m$^{-3}$), mass-weighted mean

diameter, $D_m$ (in mm), and liquid water content ($LWC$; in gm m$^{-3}$), are calculated following Bringi and

Chandrasekar (2001).

$$D_m = \frac{\int_0^\infty D^4\, N(D)\, dD}{\int_0^\infty D^3\, N(D)\, dD} \tag{10}$$

$$LWC = 10^{-3}\, \frac{\pi}{6}\, \rho_w \int_0^\infty D^3\, N(D)\, dD \tag{11}$$

$$N_w = \frac{4^4}{\pi\, \rho_w}\left(\frac{10^3\, LWC}{D_m^4}\right) \tag{12}$$

The single point-wise instrument is not sufficient to address the orographic impacts on

precipitation formation and DSD. One of the lacking to study the effect of orography is the

unavailability of many disdrometers deployed in the windward side of the WG, which could really

capture the topography variations across the WG. However, in the present study, an attempt is made to

understand the DSD difference between windward and leeward sides of the WG Mountains, to

understand the effect of orography on DSD. For this, the DSD measurements collected from Global

Precipitation Measurement (GPM) mission satellite estimates are used for the monsoon months of 2014-

2015. GPM level 3 data provides different DSD parameters like $D_m$ and $N_w$ at a spatial resolution of 0.1$^o$

× 0.1$^o$ over 60$^o$S to 60$^o$N. GPM is the first space-borne dual precipitation radar (DPR) contains Ku and



at 13.6 GHz and Ka-band at ~35.5 GHz. The details of the satellite mission can be found in Huffman et

al. (2015) and the dataset used in the present analysis can be found in Krishna et al. (2017).

Apart from this, the European Centre for Medium-Range Weather Forecasts (ECMWF) interim

reanalysis (ERA-Interim, Dee et al. 2011) dataset is also used to understand the dynamical properties

responsible for different DSD characteristics during wet and dry spells. ERA-Interim provides

atmospheric data on 60 levels in the vertical from the surface to 0.1 hPa, up to an altitude of about 80

km. ERA-Interim data are available from 1979-2019 at 3-hourly and 6-hourly intervals. The ERA-

Interim generates gridded data, including a large variety of surface parameters as well as at different

pressure levels that describe the weather as well as land surface and ocean conditions. In the present

study, temperature (K), specific humidity (kg kg$^{-1}$), and horizontal winds (m s$^{-1}$) at 700 hPa with a

spatial resolution of $0.25^{o} \times 0.25^{o}$ at 0000 UTC (LT = UTC+0530 hrs) are considered during the

monsoon seasons of 2012-2015. The specific humidity at 700 hPa infers the amount of water vapor

available for the cloud formation over the study region, WGs.

The daily accumulated rainfall collected by the India Meteorological Department (IMD) is used

to identify wet and dry spells of ISM. IMD collects the rainfall accumulations at 08:30 LT

(LT=UTC+05:30 hrs) every day.  To check the quality of JWD data, the daily accumulated rainfall

measured by JWD is compared with the daily accumulated rainfall collected from the IMD rain gauge.

For the comparison, JWD rainfall data accumulated at 08:30 LT is calculated for all the days during the

monsoon season of 2015. The daily accumulated rainfall collected by IMD and JWD above 1 mm is

considered for the comparison. A total of 76 days of data is available for the comparison. The non-

availability of data during this season may be either due to the maintenance activity or due to non-rainy





days. Figure 1 shows the scattered plot of daily accumulated rainfall for JWD and IMD collected

rainfall. A linear fit is carried out to the scatter plot and is shown with the grey line in the figure. The

correlation coefficient is found to be 0.99 between the two measurements. The bias in JWD measured

rainfall is -0.681 mm, and root mean square error is 2.875 mm. These results suggest that the JWD

measurements can be utilized to understand the DSD characteristics during wet and dry spells over

WGs.

## 3. Identification of wet and dry spells

In the present study, an objective methodology proposed by Pai et al. (2014) is used to identify

the wet and dry spells during the ISM over WGs. For this, long-term (1979-2011) high-resolution

($0.25^{o} \times 0.25^{o}$) daily gridded rainfall dataset from IMD is utilized over Mahabaleshwar ($17.75^{o}$N-$18^{o}$N

and $73.5^{o}$E-$73.75^{o}$E), WGs. The area-averaged daily rainfall time series are constructed for this region

for the monsoon period (1st June to 30th September) for the four years (2012- 2015) as well as the

monsoon period for the long-term data to identify wet and dry spells over WGs.

For a given monsoon period, the difference of the daily average rainfall for four seasons and the

daily average of the long-term data provides the daily anomalies. The standard deviation of daily

average rainfall is calculated from the long-term dataset. The standardized anomaly time series is

obtained by normalizing the daily anomalies with the corresponding standard deviations.

$$Events = \frac{(Av.of\ daily\ rain - Av.of\ long\ term\ rain)}{St.dev.of\ daily\ rain} \tag{13}$$

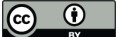

These standardized anomaly time series are used to separate the wet and dry spells in the ISM for the period 2012-2015 over WGs. A period in this standardized anomaly time series is marked as wet (dry) if the standardized anomaly exceeded a value of 0.5 (-0.5) for consecutive three days or more (Utsav et al., 2019). Figure 2 shows the standardized rainfall anomalies calculated using eq. (13). Table 1 shows the number of wet and dry days during the study period. It is observed that dry days are more during the 2012-2015 monsoon seasons, and July has comparatively more number of wet days. In this work, 44,640 (149,760) 1-min raindrop spectra are analyzed during the wet (dry) days for 2012-2015 of ISM.

## 4. Results and Discussion

The DSD and rain integral parameters during wet and dry spells are studied in terms of diurnal and with different types of precipitation (convective and stratiform) over WGs. In this study, the raindrops with diameters less than 1 mm are considered as small drops, with diameters in the range 1-4 mm are considered as mid-size drops and with diameters above 4 mm are considered as larger drops.

### 4.1. Effect of orography on raindrop size distribution over WGs

To address the DSD variations in the windward side and leeward side of the WGs, three different locations are selected; over Ocean, high altitude cloud physics laboratory (HACPL; located on the top of the WGs) and leeward side of the WGs. The DSD differences in these three sites can partially provide the effect of orography on DSD. It is to note that the satellite measurements suffer errors/shortcomings when representing the rainfall over orographic regions due to their larger footprints. Murali Krishna et al. (2017) showed that the GPM measurements improved over the Western Ghats region compared to TRMM measurements. They further assessed the DSD measurements from GPM

with ground-based measurements and found that the GPM underestimates (overestimates) the $D_m$ ($N_w$)

values over high terrain site. Hence, it is reasonable to assume that the GPM could be able to represent

the orographic features compared to other satellite measurements. However, the magnitude may not be

the same. Despite this limitation, the GPM measuremnets can be used to study the spatial differences in

DSD parameters in a statistical manner.Figure 3 shows the distribution of $D_m$ over these three locations.

In this plot, the box represents the data between first and third quartiles, and the whiskers show the data

from 12.5 and 87.5 percentiles. The horizontal line within the box represents the median value of the

distribution. It is clearly evident that the distribution of $D_m$ is narrow over the ocean and high altitude

site, whereas the $D_m$ shows large variability over the leeward side. Further, the median value of $D_m$ is

small over the ocean compared to the windward and leeward sides of the mountain. The narrow

distribution of DSD over ocean and high altitude site can be attributed to the predominance of shallow

clouds/cumulus congestus. In addition, the smaller median $D_m$ represents the shallow convection over

the ocean. Zagrodnik et al. (2019) also observed narrow DSD during the Olympic Mountains

Experiment (OLYMPEX) on the windward side of the Olympic peninsula. Similarly, the large

variability in $D_m$ on the leeward side of the mountain represents the presence of deeper clouds. These

results are consistent with the results of Utsav et al. (2017) that the congestus clouds are abundant on

the windward side and deeper clouds on the leeward side of the mountains.

### 4. 2. Diurnal variation in raindrop size distribution

The diurnal evolution of precipitation is a fundamental characteristic of regional weather

patterns. The information on the background microphysical processes, which are responsible for

precipitation formation in convective and stratiform systems, could be inferred from observed variations

in the DSDs at the ground. Figure 4 shows the temporal evolution of normalized raindrop concentration during wet and dry spells, exhibiting distinct diurnal features. It is clear that the concentration of smaller drops (Figure 4a) is higher during dry spells compared to wet spells. The presence of a large number of smaller drops in dry spells indicates the predominance of orographic convection over WGs. In the

mountain regions, DSDs evolved through warm/shallow rain processes. This warm rain is produced when the upslope wind is stronger, and moisture availability is high (White et al., 2003). In such a situation, the strong orographic wind enhances the growth of super-cooled raindrops via condensation and coalescence (Martner et al., 2008; Konwar et al., 2014). These condensational growth of raindrops produces a large number of small raindrops that are dynamically forced to form shallow rain over these

mountain regions. Further, the large number of small raindrops during dry spells indicates that the evaporation would take place during dry spells. In the smaller drop spectra, dry spells exhibit a strong diurnal cycle with primary maximum in the afternoon hours (1500-1900 LT) and secondary maximum in the night time (2300-0500 LT). This diurnal feature is also noted in Utsav et al. (2019) in the 15-dBZ echo top height (ETH) using X-band radar observations during dry spells. However, such a diurnal

cycle is not present in smaller drops during wet spells. These smaller drops shows a little higher concentration during morning hours (0500-0700 LT), representing the oceanic nature of rainfall (Krishna et al., 2016; Rao et al., 2009).

In the mid-size drops (Figure 4b), the concentration is higher in wet spells compared to dry spells. The higher concentration of mid-size drops during wet spells are due to the collision-coalescence

process (Rosenfeld and Ulbrich, 2003) and accretion of cloud water by raindrops (Zhang et al., 2008). This indicates that the congestus clouds are omnipresent during wet spells. Further, in the mid-size



drops, both the spells exhibit a diurnal cycle; however, their strengths are different. The wet spells

exhibit two broad maxima, one in late afternoon (1400-1900 LT) and the other in the early morning

(0500-0700 LT) times. The dry spells also shows two maxima, one in the late afternoon (1400-1900

LT) as in wet spells, and the other in the night time (2300-0500 LT). Such a diurnal cycle is also

observed in rainfall features over WGs (Shige et al., 2017; Romatschke and Houze, 2011). Shige et al.

(2017) observed a continuous rainfall with a double-peak structure of nocturnal and afternoon-evening

maxima over the WGs. Romatschke and Houze (2011) observed a double peak rainfall pattern in the

WGs region. They proposed that the morning peak is related to oceanic convection while the afternoon

peak is associated with the inland convection.

### *4. 3. Average raindrop spectra*

Figure 5 shows the mean DSDs during wet and dry spells along with the seasonal mean DSD

during the study period. Here $N(D)$ is plotted in a logarithmic scale to accommodate its large variability.

In general, the DSDs during dry spells are narrower than the DSDs during wet spells. The mean DSDs

are concave downward during both wet and dry spells. The mean concentration of smaller drops (< 0.9

mm) is higher, and the mean concentration of medium and larger drops is smaller in dry spells. An

increased concentration in smaller drops and a decrease in medium and larger drop concentration is

found in dry spells compared to the seasonal mean concentration. This may be due to collision and

breakup processes, as described by Rosenfeld and Ulbrich (2003) and Konwar et al. (2014). In contrast,

an increase in number concentration of drops above 0.9 mm diameter is observed in the wet spells. This

characteristic of DSD shows the wet spells has higher rainwater content and rainfall rate, which will be

discussed in the later sections. These results are distinctly different from the previous studies over other

regions, e.g., Konwar et al. (2006), Rao et al. (2009), and Harikumar et al. (2016). Harikumar et al.

(2016) observed a higher number of larger drops at three different stations located in southern India.

Konwar et al. (2006) and Rao et al. (2009) showed that stratiform precipitation dominates over

Gadanki. However, shallow convection occurs frequently over WGs (Utsav et al., 2017). This indicates

that the rain microphysical processes are different over WGs compared to other regions.

### 4. 4. Rain integral parameters during wet and dry spells

Figure 6 presents the histograms of DSD parameters, $D_m$, $\log_{10}(N_w)$, $\Lambda$, and $\mu$ during wet and dry

spells. The histograms of $D_m$ are positively skewed during both wet and dry spells (Figure 6a). The

distribution of $D_m$ is broader in dry spells compared to the wet spells. The $D_m$ value varies from 0.42 to

4.8 mm with maximum occurrence at ~1.2 mm during wet spells, whereas it varies from 0.4 to 5 mm

with maximum occurrence at ~0.8 mm during dry spells. For $D_m$ values < 1mm, the distribution for the

dry spells is higher than for the wet spells. This clearly indicates the predominance of smaller drops

during dry spells. The orography of WGs triggers the collision and breakup process in the cloud drops,

thereby producing the smaller drops (Konwar et al. 2014). The mean value of $D_m$ along with the

standard deviation and skewness, are provided in Table 2. The mean value of $D_m$ is 1.30 mm and its

standard deviation is 0.38 during wet spells, whereas the mean $D_m$ is 0.92 mm, and its standard

deviation is 0.37 during dry spells. The histograms of $\log_{10}(N_w)$ are negatively skewed during both wet

and dry spells (Figure 6b). The $\log_{10}(N_w)$ shows an inverse relation with $D_m$ and is varied from 0.52 to

5.11 during wet spells and 0.50 to 5.43 during dry spells. The histogram of the $\log_{10}(N_w)$ peak at 3.9

during wet spells. The histograms of $\log_{10}(N_w)$ shows a bimodal distribution during dry spells. This

bimodal distribution of $\log_{10}(N_w)$ peaks at 3.9 and 5. This is consistent with the results of Utsav et al.

(2019). They analyzed the 0 dBZ echo top heights, which represent the cloud top heights during wet and dry spells. They observed a bi-modal distribution in 0 dBZ echo top height, which peaks at 3 km and 6.5 km during dry spells.  The large value of standard deviation indicates the large variations in $D_m$ and $N_w$ during both wet and dry spells. The histograms of slope parameter ($\Lambda$) and shape parameter ($\mu$) during wet and dry spells are shown in Figure 6(c)-(d). The slope parameter $\Lambda$ represents the truncation of the DSD tail with the raindrop diameter. If the $\Lambda$ values are small, the DSD tail is extended to the

larger diameter and vice-versa. The shape parameter $\mu$ indicates the breadth of DSD. The positive (negative) values of $\mu$ indicates the concave downward (upward) shape for the DSD. The zero value of $\mu$ represents the exponential shape for DSD (Ulbrich, 1983). The histogram of $\Lambda$ shows positive values during both wet and dry spells. The occurrence of $\Lambda$ is higher below 10 mm$^{-1}$ during wet spells, indicating the broader spectrum of raindrops, whereas it is distributed up to 20 mm$^{-1}$ during dry spells.

The extension of $\Lambda$ towards higher values represents the higher occurrence of smaller drops during both the spells. The histogram of $\mu$ shows both positive and negative values with a higher percentage of positive values during both wet and dry spells indicating the concave downward shape of DSD during both the periods.

Overall, the DSDs over WGs are characterized by smaller $D_m$ values, and larger values of $N_w$, $\Lambda$,

and $\mu$. This represents the predominance of warm rain processes over WGs. Das et al. (2017) and Utsav et al. (2017) reported that the storms in the WGs are dominated by shallow convection. Maheskumar et al. (2014) investigated the microphysical mechanisms responsible for high rainfall over WGs and suggested that the monsoon season is characterized by low updraft speeds and low cloud liquid water content. These low updraft speeds allow sufficient time for the clouds to produce rainfall through the




warm rain process. Konwar et al. (2014) investigated the precipitating clouds over WGs induced by

orography and suggested that the collision and coalescence processes are the dominant mechanisms.

These collision and coalescence processes result in shallow convective clouds over WGs.

The variability in DSD is represented in terms of normalized distribution of $N_w$ and $LWC$ with

$D_m$ in Figure 7. The most frequent values of $N_w$ are centered near $D_m$ =1.5 mm during wet spells. The

median value of $\log_{10}(N_w)$ is 3.68 in the wet spells. Further, the moderate density of points is present

around $D_m$=1 mm and $\log_{10}(N_w)$ between 4 and 5. This is associated with the peak in $LWC$ near 1 g m$^{-3}$

at $D_m$=1 mm during wet spells. The $D_m$-$\log_{10}(N_w)$ distribution is broader in the wet spells. It is evident

from this figure that the $\log_{10}(N_w)$ shows a bimodal distribution during dry spells. The primary peak in

$\log_{10}(N_w)$ is observed at $D_m$=0.8 mm and $\log10(N_w)$>5, and the secondary peak is observed near

$\log_{10}(N_w)$=4 and $D_m$=1 mm. Even though it is presented, this bimodal distribution is weak in wet spells.

The primary peak in $\log_{10}(N_w)$ corresponds to the peak in $LWC$. The $LWC$ values during dry spells

peaks at lower $D_m$ (<1mm) compared to $LWC$ peak during wet spells. The distribution of $LWC$ reveals

that both wet and dry spells are characterized by low $LWC$, suggesting the dominance of shallow

convection over WGs. The $LWC$ shows a higher occurrence below 0.3 gm$^{-3}$ and 0.1 gm$^{-3}$ during the wet

and dry spells, respectively. Das et al. (2017) studied the DSD spectra for different types of

precipitation and revealed that the shallow clouds are characterized by low $LWC$ (<0.5 gm$^{-3}$) over WGs.

Further, the distribution of $LWC$ is narrower in dry spells compared to wet spells. It is apparent from the

above discussion that the wet spells are characterized by larger $D_m$, smaller $N_w$, and high $LWC$, whereas

the dry spells are characterized by smaller $D_m$, larger $N_w$, and low $LWC$.



To study the differences in DSD during wet and dry spells with rainfall rate, the distribution of $N(D)$ is compared at different rain rates in Figure 8. Here $N(D)$ is plotted in logarithmic scale. It is evident from this figure that significant differences exist in $N(D)$ from wet to dry spells. The contours are shifted to higher rain rates and higher diameters in wet spells compared to dry spells. This indicates that the mid-size drops in the range 1-2 mm are higher in wet spells than in dry spells for the same rain

rate. This is more pronounced in lower rain rates below 10 mm hr$^{-1}$. At higher rain rates (above 10 mm hr$^{-1}$), the smaller and mid-size drops are higher in wet spells than in dry spells. However, this difference decreases gradually as rain rate increases. At above 30 mm hr$^{-1}$, both the spells show a similar distribution of $N(D)$ (not shown in the figure). However, in the larger drop diameters above 4.5 mm, the concentration is higher in the wet spells compared to dry spells in all the rain rate intervals.

The mass-weighted mean diameter, $D_m$, and intercept parameter, $N_w$, represent the characteristics of overall DSD features. Previous observational studies showed that both $D_m$ and $N_w$ vary with rain type and intensity (e.g., Thurai et al., 2010; Marzano et al., 2010; Chen et al., 2013; Wen et al., 2016). To investigate the dependence of these parameters, $D_m$ and $N_w$ on rainfall intensity, the normalized occurrence of $D_m$-R and $N_w$-R, along with the fitted power-law relation, is shown in Figure 9 for both

wet and dry spells. The $D_m$ and $N_w$ show a nearly linear relation with rainfall intensity. The occurrence of smaller Dm is higher for lower rain rates in both the spells. Among the two spells, the dry spell shows a higher occurrence below 1 mm. An inverse relationship with a smaller frequency of Nw is evident in wet spells. It can be seen that the coefficient and exponent in both $D_m$-R and $N_w$-R relations are positive, indicating the enhancement in $D_m$ and $N_w$ values with the increase in rain intensity during

both wet and dry spells. The increase in $D_m$ and $N_w$ values with rain rate is due to the efficient collision

coalescence and breakup processes (Chen et al., 2013). It is evident from the figure that the distribution

is significantly broader in low rain rates. This indicates the higher variability in DSD at lower rain rates.

It is interesting to note here that the $D_m$ and $N_w$ values reach an equilibrium state at higher rain rates.

The general process to attain the equilibrium DSD is by the collision, coalescence and breakup

mechanisms as described in Hu and Srivastava (1995) and elaborated in Atlas and Ulbrich (2000).

Under this equilibrium condition, $D_m$ is constant and a further increase in rain rate is due to the increase

in concentration. It is observed here that the $D_m$ value approaches 2-2.1 mm when rain intensity

increases above 40 mm hr$^{-1}$ in both wet and dry spells, which indicates the equilibrium state of the

DSDs. The coefficient value in $D_m$-$R$ relation is comparably higher in the wet spells. This indicates that,

for a given rain rate, the $D_m$ values are higher in wet spells than in dry spells. This may be due to

collision and coalescence processes (Rosenfeld and Ulbrich, 2003) and the accretion of cloud water by

raindrops (Zhang et al., 2008). The $N_w$-$R$ shows an inverse relationship such that the dry spells show a

higher coefficient compared to wet spells. This indicates that the concentration of smaller drops is

higher in dry spells compared to wet spells, which is due to collision, breakup and evaporation

processes, as explained by Rosenfeld and Ulbrich (2003).

The above results indicate that the rainfall over WGs is associated with warm rain processes

during both wet and dry spells. The microphysical processes in warm rain include rain evaporation,

accretion of cloud water by raindrops and rain sedimentation, etc. (Zhang et al., 2008). Giangrande et al.

(2017) observed the predominance of larger cloud droplets in warm clouds during the wet spells over

Amazon. Similarly, Machado et al. (2018) showed that the larger $D_m$ values are associated with the

mixed-phase clouds during dry spells over Amazon. Recently, using X-band radar data, Utsav et al.



(2019) showed that 0 dBZ echo top height (ETH, represent cloud top height) peaks at about 3 km during dry spells, whereas during the wet spell, 0 dBZ ETH peaks at about 5-7 km (indicating the presence of cumulus congestus). Thus, the larger values of $D_m$ may be due to the presence of cumulus congestus

during wet spells. To understand the dynamical mechanisms leading to different microphysical processes during wet and dry spells, we have analysed temperature, specific humidity, and horizontal winds during the monsoon seasons of 2012-2015 over WGs. The temperature anomalies at 700 hPa derived from the ERA-Interim reanalysis dataset during wet and dry spells are shown in Figure 10. This level is chosen, as the temperature anomaly and the availability of moisture at this level aid the growth

of active convection. It is observed that the temperature is cooler over the west coast of India (including the study region) in wet spell compared to that in dry spell. The higher temperature in the dry spell can lead to the evaporation of raindrops, which subsequently can break the drops thereby leading to lesser diameter drops in dry compared to the wet spell. Figure 11 shows the mean specific humidity (kg kg$^{-1}$) and mean horizontal velocity (m s$^{-1}$) at 700 hPa derived from the ERA-Interim reanalysis dataset. In this

plot, the color bar represents mean specific humidity. It is observed that the specific humidity is higher over WGs during wet spells compared to dry spells. The thermal gradient between WGs and surrounding regions, and the availability of more moisture favors the growth of active convection in the wet spell compared to that of dry spell. It is known that the vertical velocity during the wet spell is stronger compared to the dry spell (Uma et al., 2012). The strong updrafts aid the growth of cloud liquid

water particles and thereby increase the size of the drops.

Further, the mean wind pattern represents that the winds are of marine nature, originating from the Arabian Sea in both the spells; however, their magnitude is different. These oceanic winds transport



the sea salt aerosols from the Arabian Sea as well as dust from the local sources, which acts as giant

cloud condensation nuclei (CCN) during moist conditions. These giant CCN are responsible for warm

rain processes over WGs (Kumar et al., 2013). Aerosols can affect the formation of cloud droplets in

terms of their size and number concentration (Twomey, 1977; Twomey et al., 1984). For a given

amount of CCN, the amount of water vapour available for the formation of congestus clouds is higher

during wet spells, which results in comparatively larger drops during this period. Hence, the higher

availability of water vapour, strong horizontal winds and giant CCN favours for the formation of

cumulus congestus. These congestus are responsible for the presence of medium size/larger drops

during wet spells.

In addition, the raindrop diameter depends on the rain rate, which varies between wet and dry

spells. The distribution of $D_m$ during wet and dry spells at different rain rates are shown in Figure 12.

For lower rain rates (below 10 mm hr$^{-1}$), the raindrops falling from the cloud tops can grow by

deposition of water vapour and accretion of cloud water during wet spells. This results in larger $D_m$

values during wet spells compared to dry spells. At higher rain rates (above 20 mm hr$^{-1}$), the $D_m$

distribution remains the same during both the spells. This is due to the equilibrium of DSD by the

collision, coalescence, and breakup mechanisms, as described in Hu and Srivastava (1995) and Atlas

and Ulbrich (2000). Hence, it is evidenced that even though warm rain is predominant during both wet

and dry spells, the different dynamical mechanisms lead to different DSD characteristics during wet and

dry spells.

*4. 5. Raindrop size distribution during stratiform and convective regimes*





Numerous studies have been carried out to understand the DSDs during different storms and within a storm (Dolman et al., 2011; Munchak et al., 2012; Friedrich et al., 2013; Thompson et al., 2015; Dolan et al., 2018). This can be attributed to combined dynamical (stratiform and convective) and microphysical processes occurring in the storms. To understand the effect of dynamical processes on observed variations in DSD during wet and dry spells, the observed precipitation events are classified into stratiform and convective types based on the method proposed by Bringi et al. (2003). In their study, they considered 5 consecutive 2 min DSD samples to classify precipitation into stratiform and convective types. However, in the present study, 10 consecutive 1 min DSD samples are considered, and further, if the standard deviation of 10 consecutive DSD samples is less than 1.5 mm hr$^{-1}$, then the precipitation is classified as stratiform; otherwise, it is classified as convective.

Figure 13 presents the histograms of $D_m$, $\log_{10}(N_w)$, $\Lambda$, and $\mu$ during stratiform rain events in wet and dry spells. The mean, standard deviation, and skewness of these parameters are provided in Table 3. The histograms of $D_m$ (Figure 13a) are positively skewed during stratiform rain events in both wet and dry spells. The histogram of $D_m$ is broader in dry spells compared to wet spells. The value of $D_m$ varies from 0.42 to 2.48 mm with a maximum near 1-1.4 mm during stratiform rain in wet spells. Whereas, it varies between 0.38 and 2.77 mm with maximum occurrence near 0.42-0.58 mm during stratiform rain in dry spells. The distribution of $D_m$ shows a higher frequency below 0.6 mm in dry spells. This indicates that the presence of more number of smaller raindrops in stratiform rain of dry spells compared to wet spells. Similarly, the distribution of $D_m$ is higher in wet spells above 1 mm, indicating the dominance of medium size and/or larger drops in stratiform rain of wet spells compared to dry spells. The histogram of $\log_{10}(N_w)$ (Figure 13b) is positively skewed in stratiform rain in wet spells and

negatively skewed in stratiform rain in dry spells. The distribution is narrower in wet spells and broader

in dry spells. The distribution peaks between 3-3.6 during wet spells, whereas it peaks at 5 during dry

spells. The distribution of $\Lambda$ (Figure 13c) is broader in the stratiform rain events during both wet and dry

spells. The distribution varies from 1.2 mm$^{-1}$ to 52 mm$^{-1}$ with a mode at 10 mm$^{-1}$ in the stratiform rain of

wet spells. This further supports the presence of mid-size drops during wet spells. The distribution of $\Lambda$

shows higher occurrences above 15 mm$^{-1}$ during dry spells, indicating the truncation of DSD at

relatively smaller drop diameters compared to wet spells. The histograms of $\mu$ (Figure 13d) indicate a

concave downward shape for DSDs during stratiform rain events in both wet and dry spells.

Figure 14 shows the distribution of $D_m$, $\log_{10}(N_w)$, $\Lambda$, and $\mu$ during convective rain events in wet

and dry spells. The histograms of $D_m$ are positively skewed in convective rain during both wet and dry

spell (Figure 14a). In convective rain, the distribution of $D_m$ is broader in wet spells compared to dry

spells. It can be clearly seen that the presence of small drops is higher in dry spells compared to wet

spells even in convective rain also. The distribution of $\log_{10}(N_w)$ shows an inverse relation with $D_m$ in

convective rain (Figure 14b). The $\log_{10}(N_w)$ is negatively skewed in wet spells, whereas it is positively

skewed in dry spells. The distribution of $\Lambda$ (Figure 14c) indicates the presence of larger drops in

convective rain compared to stratiform rain in both wet and dry spells. The histograms of $\mu$ (Figure 14d)

indicate the concave downward shape of DSDs in convective rain of both wet and dry spells. The mean,

standard deviation, and skewness of these parameters are provided in Table 4.

Several points can be noted from the above discussion:

1. The maximum value for mean $D_m$ and the largest standard deviation are found for convective rain in

wet spells.

2. The maximum value for $\log_{10}(N_w)$ and higher standard deviation are observed during stratiform rain

in dry spells.

3. An appreciable difference is found in the histograms of $D_m$ and $\log_{10}(Nw)$ during the stratiform rain in

dry and wet spells. However, this difference is small in the convective rain.

4. Even in the histograms of $\Lambda$ and $\mu$, the distinct differences exist in stratiform rain during wet and dry

spells.

Figure 15 presents the scatter plot of $D_m$ and $\log_{10}(N_w)$ during the stratiform and convective rain

in wet and dry spells, as well as the statistical results reported by Bringi et al. (2003). The dashed line

corresponds to the stratiform line, and two gray rectangles represent the maritime and continental

clusters reported by Bringi et al. (2003). It is evident from this plot that there exist two distinct

precipitation groups that are well separated in $D_m$-$N_w$ space, corresponding to convective and stratiform

rain. Some of the convective data points in our study appear in the maritime cluster, and a few points

are present in the continental clusters as shown by Bringi et al. (2003). This indicates that the

precipitation over WGs is different from Bringi et al. (2003), even though the precipitation over WGs

resembles maritime nature. The stratiform rain has a wide variability ranging from 0.37 to 2.8 mm for

$D_m$ and from 1.3 to 5.5 for $\log_{10}(N_w)$ and is plotted below the stratiform line of Bringi et al. (2003). The

convective rain has relatively lower variability in both wet and dry spells. Among the two spells, the

convective rain in dry spell has the lowest variability in $D_m$ and $\log_{10}(N_w)$. Figure 16 shows the average

values of $D_m$ and $\log10(N_w)$ along with standard deviation for the stratiform and convective rain during

wet and dry spells as well as the statistical results reported by Bringi et al. (2003). It is evident here that

an inverse relationship between $D_m$ and $\log_{10}(N_w)$ during stratiform and convective rain. The mean $N_w$

values in our study are higher than the Marshall-Palmer value of $\log_{10}(N_w)$ (3.9) for an exponential

shape of DSD. The stratiform rain has smaller $D_m$ and larger $N_w$. The convective rain appears at $\langle D_m \rangle \approx$

1.5-1.75 and $\log_{10}(N_w) \approx$ 3.9-4.3. The convective rain in our study falls near maritime convective

clusters of Ulbrich and Atlas (2007).

### *4. 6 μ-Λ relation*

The gamma distribution function has been widely used in the microphysical parameterization

schemes in the atmospheric models to describe various DSDs. However, $\mu$ is often considered to be

constant. Milbrandt and Yau (2005) found that $\mu$ plays an important role in determining sedimentation

and microphysical growth rates. In this context, the microphysical properties of clouds and precipitation

are sensitive to variations in $\mu$. Several researchers showed that the value of $\mu$ varies during the

precipitation (Ulbrich, 1983; Ulbrich and Atlas, 1998; Testud et al., 2001; Zhang et al., 2001; Islam et

al., 2012). Zhang et al. (2001) proposed an empirical relation between $\mu$ and $\Lambda$ to retrieve the gamma

distribution. Zhang et al. (2003) proposed an empirical $\mu$-$\Lambda$ relation using 2D video disdrometer data

collected in Florida. They examined the $\mu$-$\Lambda$ relation with different types of precipitation. These $\mu$-$\Lambda$

relations are useful in reducing the bias in rain parameters from remote measurements (Zhang et al.,

2003). Recent studies have demonstrated the variability in μ-Λ relation in different types of rain and at

different geographical locations (Chang et al., 2009; Kumar et al., 2011; Wen et al., 2016). Hence, it is

necessary to derive different $\mu$-$\Lambda$ relations based on local DSD observations, in particular, over the

orographic precipitation.

In the present study, an empirical $\mu$-$\Lambda$ relationship is derived for both wet and dry spells. To

minimize the sampling errors, the $\mu$-$\Lambda$ relation was established by filtering out the DSDs with a rainfall





rate less than 5 mm hr$^{-1}$, and total drop counts less than 1000, as proposed by Zhang et al. (2003).

Figure 17 shows the $\mu$-$\Lambda$ relation for the wet and dry spells and the corresponding polynomial least-square fits are shown as solid lines. The fitted $\mu$-$\Lambda$ relations for wet and dry spells are given as follows:

Wet spell:  $\qquad\qquad\qquad \Lambda = 0.0359\mu^2 + 0.802\mu + 2.22 \qquad\qquad$ (14)

Dry spell:  $\qquad\qquad\qquad \Lambda = 0.0138\mu^2 + 1.151\mu + 1.198 \qquad\qquad$ (15)

A similar behavior is observed for both wet and dry spells, the smaller the value of $\Lambda$ (higher rain rates), smaller is the value of $\mu$. This indicates that the DSDs tend to be more concave downwards with the increase in rainfall intensity. This suggests a higher fraction of small and mid-size drops and 535 lower fraction of larger drops, reflecting less evaporation of smaller drops and more drop breakup processes. However, the fitted $\mu$-$\Lambda$ relation exhibits a large difference for wet and dry spells. Comparing Eq. (14) and (15), one can observe that the coefficient of the linear term is smaller in wet spells than that of dry spells. Hence, for a given value of $\mu$, dry spells have a higher value of $\Lambda$ compared to the wet spells. This suggests that the single $\mu$-$\Lambda$ relation cannot reliably represent the observed phenomenon 540 during different phases of the monsoon.

Comparing the $\mu$-$\Lambda$ relations in this study with that obtained from Zhang et al. (2003), the $\mu$-$\Lambda$ relationship of the dry spell has a smaller slope. These differences reveal that the DSD during dry spell have lower values of $D_m$. This indicates that the underlying microphysical processes in the orographic precipitating systems are different from those observed over Florida during the summer of 1998. 545 Further, the $\mu$-$\Lambda$ relationships are derived for convective and stratiform rain for the JWD measurements and are provided in Figure 18. The least-square polynomial fit for convective and stratiform rain is as follows:





Convective rain:  $\qquad \Lambda = 0.0069\mu^2 + 0.576\mu + 2.42 \qquad$ (16)

Stratiform rain:  $\qquad \Lambda = 0.0022\mu^2 + 0.933\mu + 1.86 \qquad$ (17)

It is observed that the coefficients of the squared and linear term of convective precipitation are

smaller than those given by Zhang et al. (2003). Hence, for a given value of $\mu$, the convective

precipitation in the present study gives lower values of $\Lambda$ than that for the convective precipitation from

Zhang et al. (2003).

Seela et al. (2018) fitted $\mu$-$\Lambda$ relations for summer and winter rainfall over North Taiwan. Chen

et al. (2017) have derived an empirical $\mu$-$\Lambda$ relation over Tibetan Plateau. Cao et al. (2008) analyzed the

$\mu$-$\Lambda$ relations over Oklahoma. Different $\mu$-$\Lambda$ relations are derived for different weather systems over

North Taiwan (Chu and Su 2008). The $\mu$-$\Lambda$ relations derived for the present study are compared with the

other orographic precipitations on the globe and are provided in Table 5. This shows that $\mu$-$\Lambda$ relations

vary in different types of precipitation and climatic regimes.


**5. Summary**

The raindrop spectra measured by Joss-Waldvogel Disdrometer (JWD) are analyzed to

understand the DSD variations during wet and dry spells of the Indian summer monsoon over the WGs.

Observational results indicate that the mean DSDs are considerably different during wet and dry spells.

In addition, the DSD variability is studied in relation to stratiform and convective rain during wet and

dry spells.

- Overall, a high concentration of smaller drops is observed in both wet and dry spells, indicating

    the dominance of shallow convection, generally observed in orographic precipitation.



- The diurnal DSD variation shows that the concentration of smaller drops is higher in dry spells, while the concentration of mid-size drops is higher in wet spells throughout the day.

- The dry spells exhibit a strong diurnal cycle with double-peak during late afternoon and night time in both smaller and mid-size drops. Whereas, this diurnal cycle is weak for smaller drops in wet spells.

- The dry spells show a high concentration of smaller drops resulting in collision and breakup processes, as described by Rosenfeld and Ulbrich (2003).

- The higher concentration of mid-size and larger drops is observed in wet spells compared to dry spells. Higher availability of water vapour, strong horizontal winds, and giant CCN favors the formation of cumulus congestus, which are responsible for the presence of medium size/larger drops during wet spells.

- The DSDs over WGs are characterized by small $D_m$, large $N_w$. The $N_w$ shows a bi-modal distribution during dry spells. This bimodality is weakly evident in the wet spells.

- The distribution of $\Lambda$ shows the dominance of small drops in dry spells compared to wet spells and the dominance of mid-size drops in wet spells compared to dry spells. The distribution of $\mu$ represents the concave downward shape of DSDs for both wet and dry spells.

- The distribution of $N_w$ with rainfall rate represents the dominance of mid-size drops in wet spells compared to dry spells at the same rain rate.

- An empirical relation is derived between $\mu$ and $\Lambda$ during wet and dry spells. The fitted $\mu$-$\Lambda$ relation for both spells exhibits a large difference between them. The $\mu$-$\Lambda$ relation indicates the higher fraction of small and mid-size drops and the lower fraction of larger drops over the WGs.

• An appreciable difference in raindrop size distribution is observed in the stratiform rain of wet and dry spells. Higher amounts of smaller drops are evident in both stratiform and convective rain of dry spells compared to wet spells.

The presence of small drops is evident in stratiform rain, whereas larger drops are present in convective rain, as observed in earlier studies. It is evident from this study that, even though the warm

rain is predominant during both wet and dry spells, the dynamical mechanisms underlying the microphysical processes are different, which causes the difference in observed DSD characteristics during wet and dry spells. The distinct features of DSD during the wet and dry spells of the ISM over WGs are summarized in Figure 19.

**Author contributions:**

UVMK and SKD designed, analysed, and prepared the manuscript. SKD, UVMK, and UB proposed the methodology. GSE, SMD, and GP contributed with discussion to the manuscript.

**Acknowledgments:**

The authors are thankful to the Director, IITM, for his support. The authors would like to acknowledge the technical/administrative staff of the High Altitude Cloud Physics Laboratory (HAPCL), Mahabaleshwar, for maintaining disdrometer. The authors acknowledge the India Meteorological Department (IMD) for the provision of the rainfall dataset. The authors also acknowledge the JAXA, JAPAN, and NASA, USA, for the provision of the GPM data
(https://pmm.nasa.gov/data-access/downloads/gpm). The authors would like to acknowledge the





European Centre for Medium-Range Weather Forecasts (ECMWF) for the provision of the ERA-Interim dataset. The disdrometer data are archived at IITM and are available with the corresponding author (skd_ncu@yahoo.com) for research collaboration. The manuscript benefitted from comments and suggestions provided by the Editor and two anonymous reviewers.

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





**Table Captions:**

**Table 1:** Total number of wet and dry days during the monsoon seasons (June-September) of 2012 - 2015.

**Table 2:** Mean, Standard deviation, and Skewness of the DSD parameters in wet and dry spells.

**Table 3:** Mean, Standard deviation, and Skewness of the DSD parameters in stratiform rain during wet and dry spells.

**Table 4:** Mean, Standard deviation, and Skewness of the DSD parameters in convective rain during wet and dry spells.


**Figure Captions:**

**Fig.1:** Scatterplot between daily accumulated rainfall collected by IMD rain gauge and measured by JWD. The solid grey line indicates the linear regression.

**Fig 2:** The standardized rainfall anomaly for the year (a) 2012, (b) 2013, (c) 2014, and (d) 2015 during
the period June-September. Dashed line marked for 0.5 (+ve X-axis) and -0.5 (-ve X-axis) rainfall anomaly.

**Fig 3:** Box and whisker plot of $D_m$ distributions over Ocean, windward (HACPL), and leeward side of the mountain obtained from GPM measurements. Here box represents the data between first and third quartiles, and the whiskers show the data from 12.5 and 87.5 percentiles. The horizontal
line within the box represents the median value of the distribution.

**Fig 4:** Diurnal variation in raindrop concentration during wet and dry spells for (a) smaller drops (< 1mm) and (b) mid-size drops (1-4 mm). In this plot, the concentration of raindrops within each





hour are normalized with the total concentration of raindrops. Here, the black line represents wet spells, and the red line represents dry spells.

**Fig 5:** Average DSDs during wet and dry spells.

**Fig 6:** Histograms of $D_m$, $\log_{10}(N_w)$, $\Lambda$ and $\mu$ during wet and dry spells. Blackline represents wet spells, and the red line represents dry spells.

**Fig 7:** Normalized frequency of occurrence for (a)-(b) $D_m$-$\log_{10}(N_w)$ and (c)-(d) $D_m$-$LWC$.

**Fig 8:** The variation in $N(D)$ as a function of $D$ at different $R$ for (a) wet and (b) dry spells.

**Fig 9:** Normalized frequency of occurrence for (a)-(b) $D_m$-$R$ and (c)-(d) $\log_{10}(N_w)$-$R$.

**Fig 10:** Spatial distribution of temperature anomalies (K) at 700 hPa during (a) wet and (b) dry spells of the monsoon seasons of 2012-2015. Here positive anomaly represents heating, and negative anomaly represents cooling. The red dot represents the observational site.

**Fig 11:** Spatial distribution of mean horizontal winds (m s$^{-1}$) and mean specific humidity (kg kg$^{-1}$) during (a) wet and (b) dry spells of the monsoon seasons of 2012-2015. Here color bar represents the specific humidity, and arrows represent horizontal wind velocity. The black dot represents the observational site.

**Fig 12:** Distribution of $D_m$ at different rain rates during wet and dry spells. In this plot, the horizontal line within the box represents the median value. The boxes represent data between first and third quartiles, and the whiskers show data from 12.5 to 87.5 percentiles. Here black color represents wet spells, and the red color represents dry spells.

**Fig 13:** Histograms of $D_m$, $\log_{10}(N_w)$, $\Lambda$ and $\mu$ in stratiform rain during wet and dry spells. The black dashed line represents wet spells, and the red dashed line represents dry spells.





**Fig 14:** Histograms of $D_m$, $\log_{10}(N_w)$, $\Lambda$ and $\mu$ in convective rain during wet and dry spells. Blackline
920        represents wet spells, and the red line represents dry spells.

**Fig 15:** Scatterplot of $D_m$ versus $\log_{10}(N_w)$ for convective (red dots) and stratiform (black dots) rain
types in (a) wet spells and (b) dry spells. The dashed line represents the stratiform line, and two
gray rectangles represent maritime and continental clusters, as reported in Bringi et al. (2003).
The equation separating the stratiform and convective cluster used by Bringi et al. (2003) is also
925        provided in the figure.

**Fig 16:** The average value of $\log_{10}(N_w)$ and $D_m$ along with standard deviations during stratiform and
convective rain in wet and dry spells. The dashed line represents the stratiform line, and the
rectangle boxes represent maritime and continental clusters, according to *Bringi et al*. (2003).
The black (red) color represents the wet (dry) spells. The filled circles (squares) represent the
930        convective (stratiform) rain types.

**Fig 17:** Scatter plots of $\Lambda$-$\mu$ values obtained from gamma DSD for (a) wet and (b) dry spells. The solid
line indicates the least square polynomial fit for $\Lambda$-$\mu$ relation.

**Fig 18:** Scatter plots of $\Lambda$-$\mu$ values obtained from gamma DSD for (a) convective and (b) stratiform
rain. The solid line indicates the least square polynomial fit for $\Lambda$-$\mu$ relation.

**Fig 19:** Summary of the DSD characteristics during the wet and dry spells over WGs.





**Table 1: Total number of wet and dry days during the monsoon seasons (June-September) of 2012 - 2015.**

| Months | Wet (No. of. days) | Dry (No. of. days) |
|---|---|---|
| June | 15 | 40 |
| July | 16 | 38 |
| August | 0 | 46 |
| September | 10 | 35 |

**Table 2: Mean, Standard deviation, and Skewness of the DSD parameters in wet and dry spells.**

| | Wet | | | Dry | | |
|---|---|---|---|---|---|---|
| | **Mean** | **Standard deviation** | **Skewness** | **Mean** | **Standard deviation** | **Skewness** |
| $D_m$ | 1.30 | 0.38 | 0.564 | 0.92 | 0.366 | 1.412 |
| $\log_{10}(N_w)$ | 3.62 | 0.51 | -0.515 | 4.46 | 0.681 | -0.234 |
| $\Lambda$ | 15.42 | 10.25 | 1.172 | 22.01 | 12.428 | 0.478 |
| $\mu$ | 14.4 | 9.937 | 1.087 | 17.8 | 11.021 | 0.701 |
| R | 6.62 | 9.75 | 3.19 | 2.79 | 5.02 | 4.59 |
| LWC | 0.34 | 0.42 | 2.54 | 0.18 | 0.24 | 3.22 |






**Table 3: Mean, Standard deviation, and Skewness of the DSD parameters in stratiform rain during wet and dry spells.**

|  | Wet spells | | | Dry spells | | |
|---|---|---|---|---|---|---|
|  | Mean | Standard deviation | Skewness | Mean | Standard deviation | Skewness |
| $D_m$ | 1.18 | 0.314 | 0.143 | 0.75 | 0.265 | 1.284 |
| $\log_{10}(N_w)$ | 3.52 | 0.555 | 0.186 | 4.39 | 0.6792 | -0.686 |
| $\Lambda$ | 17.08 | 10.558 | 0.962 | 26.77 | 12.477 | 0.612 |
| $\mu$ | 15.12 | 10.168 | 1.016 | 20.81 | 10.758 | 0.399 |


**Table 4: Mean, Standard deviation, and Skewness of the DSD parameters in convective rain during wet and dry spells.**

|  | Wet spells | | | Dry spells | | |
|---|---|---|---|---|---|---|
|  | Mean | Standard deviation | Skewness | Mean | Standard deviation | Skewness |
| $D_m$ | 1.66 | 0.289 | 0.878 | 1.47 | 0.303 | 0.344 |
| $\log_{10}(N_w)$ | 3.86 | 0.227 | -0.538 | 4.01 | 0.292 | 0.189 |
| $\Lambda$ | 10.08 | 5.216 | 1.292 | 13.15 | 7.494 | 1.094 |
| $\mu$ | 11.86 | 6.695 | 0.774 | 14.05 | 8.732 | 1.157 |



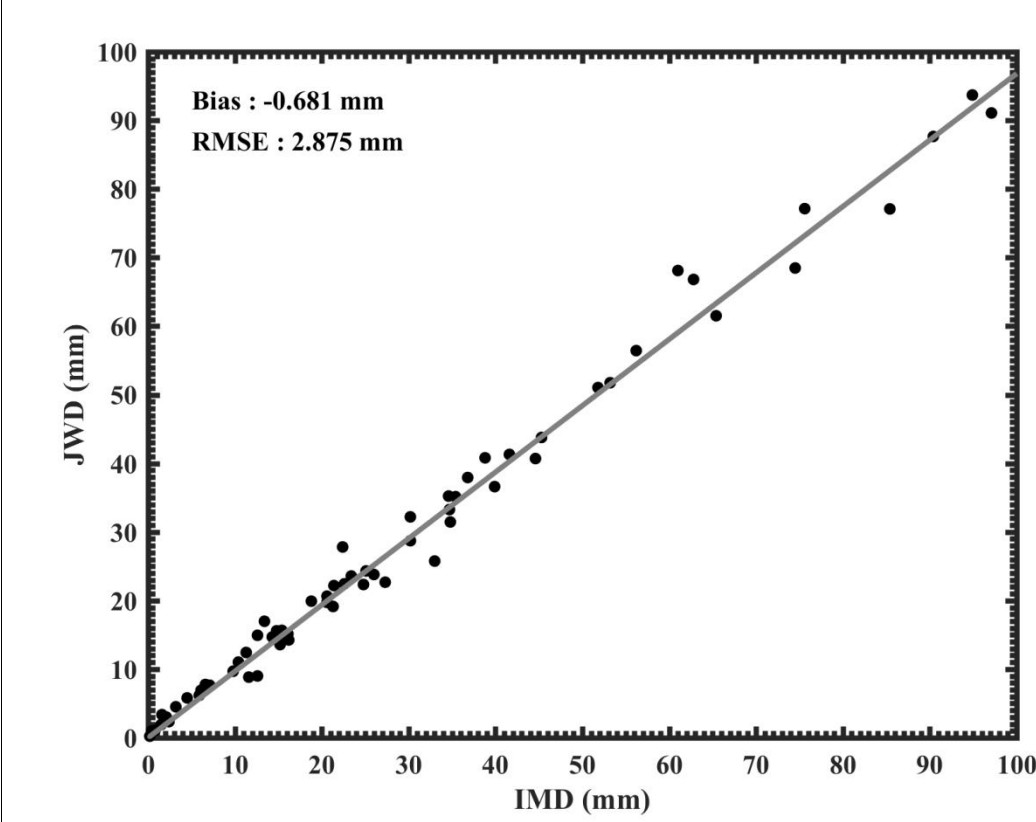

**Fig.1:** Scatterplot between daily accumulated rainfall collected by IMD rain gauge and measured by

JWD. The solid grey line indicates the linear regression.







**Fig 2:** The standardized rainfall anomaly for the year (a) 2012, (b) 2013, (c) 2014, and (d) 2015 during

the period June-September. Dashed line marked for 0.5 (+ve X-axis) and -0.5 (-ve X-axis)

rainfall anomaly.





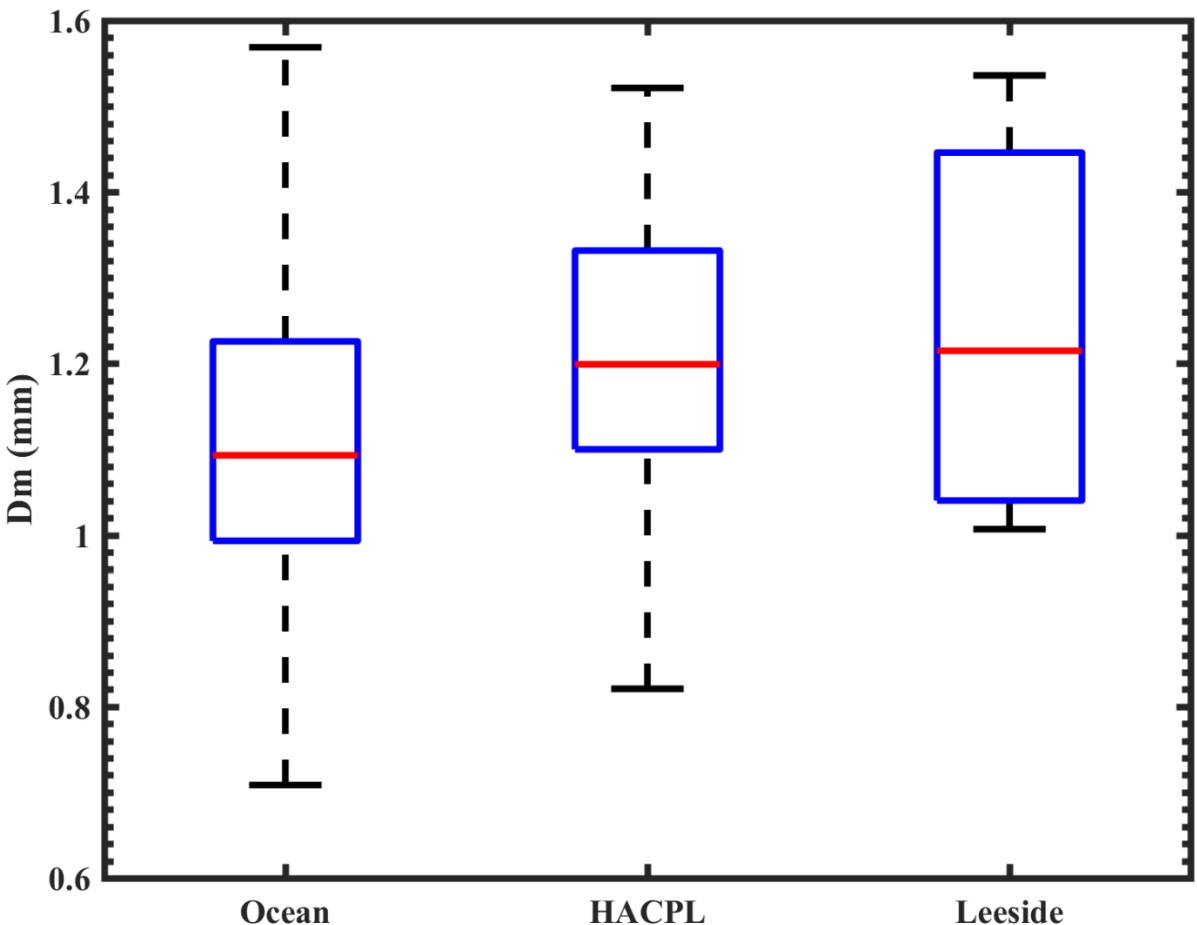

**Fig 3:** Box and whisker plot of $D_m$ distributions over ocean, windward (HACPL), and leeward side of the mountain obtained from GPM measurements. Here box represents the data between first and third quartiles, and the whiskers show the data from 12.5 and 87.5 percentiles. The horizontal line within the box represents the median value of the distribution.





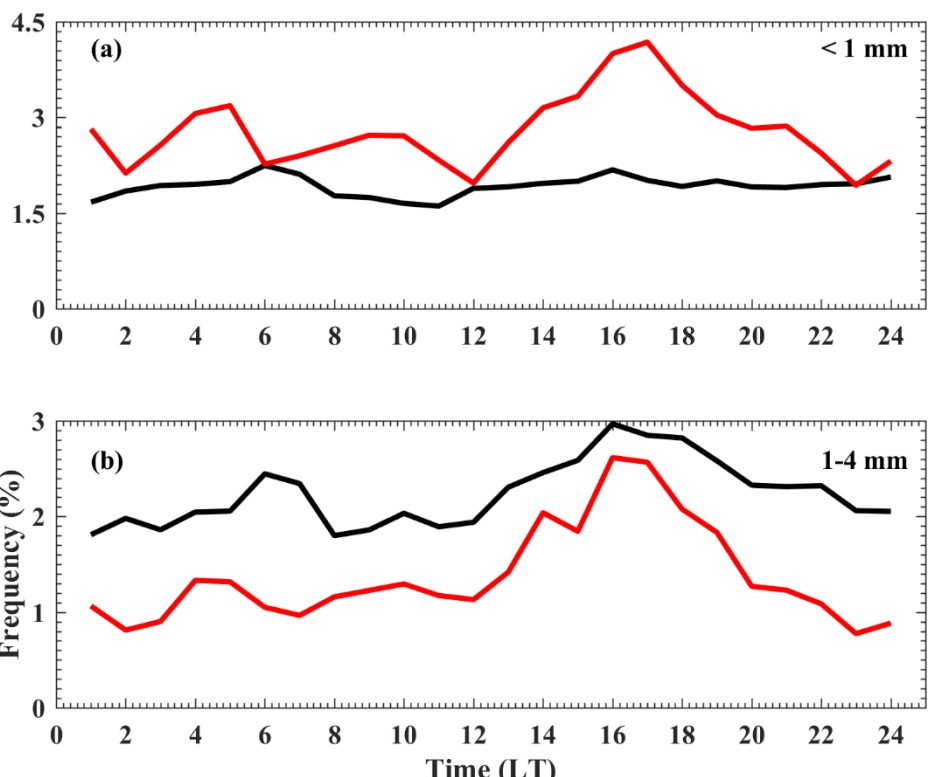

**Fig 4:** Diurnal variation in raindrop concentration during wet and dry spells for (a) smaller drops (<

1mm) and (b) mid-size drops (1-4 mm). In this plot, the concentration of raindrops within each

hour are normalized with the total concentration of raindrops. Here, the black line represents wet

spells, and the red line represents dry spells.





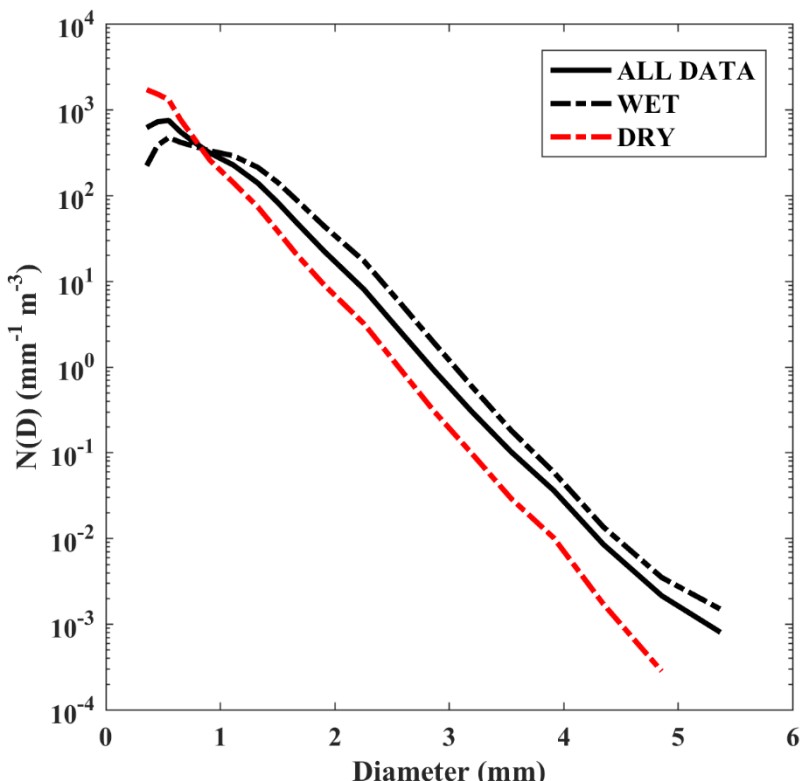

**Fig 5:** Average DSDs during wet and dry spells.





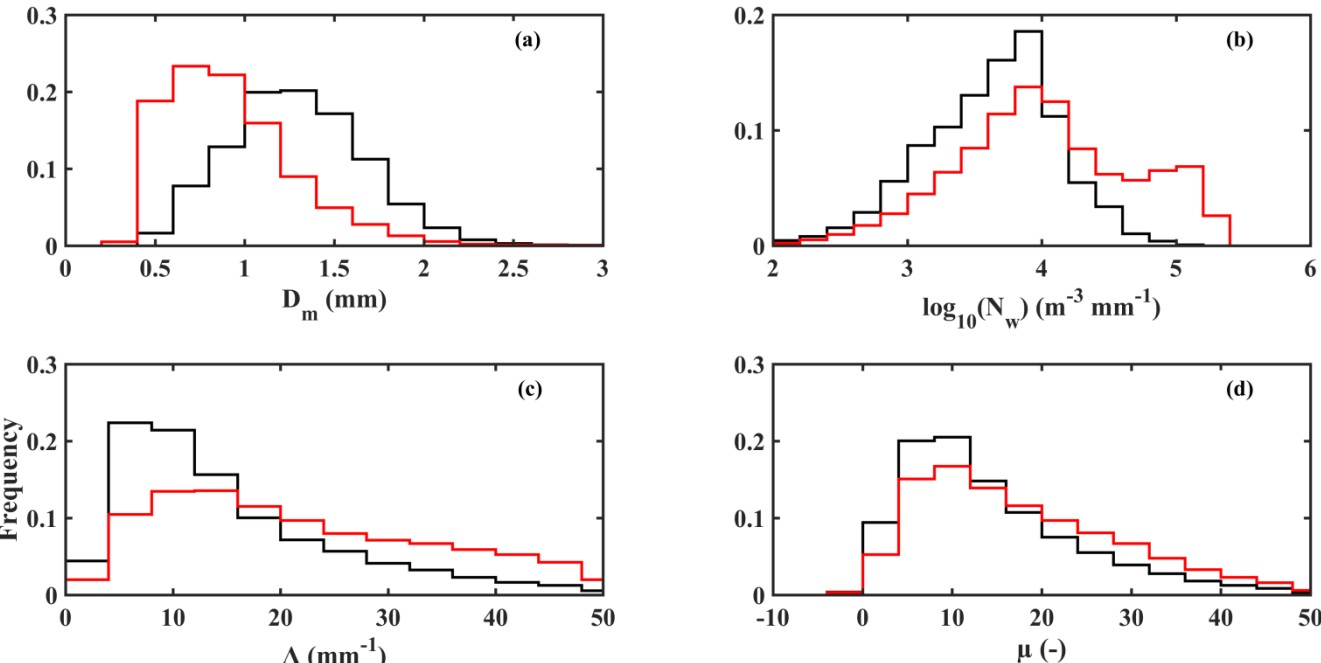

**Fig 6:** Histograms of $D_m$, log10($N_w$), $\Lambda$ and $\mu$ during wet and dry spells. Blackline represents wet spells,

and the red line represents dry spells.





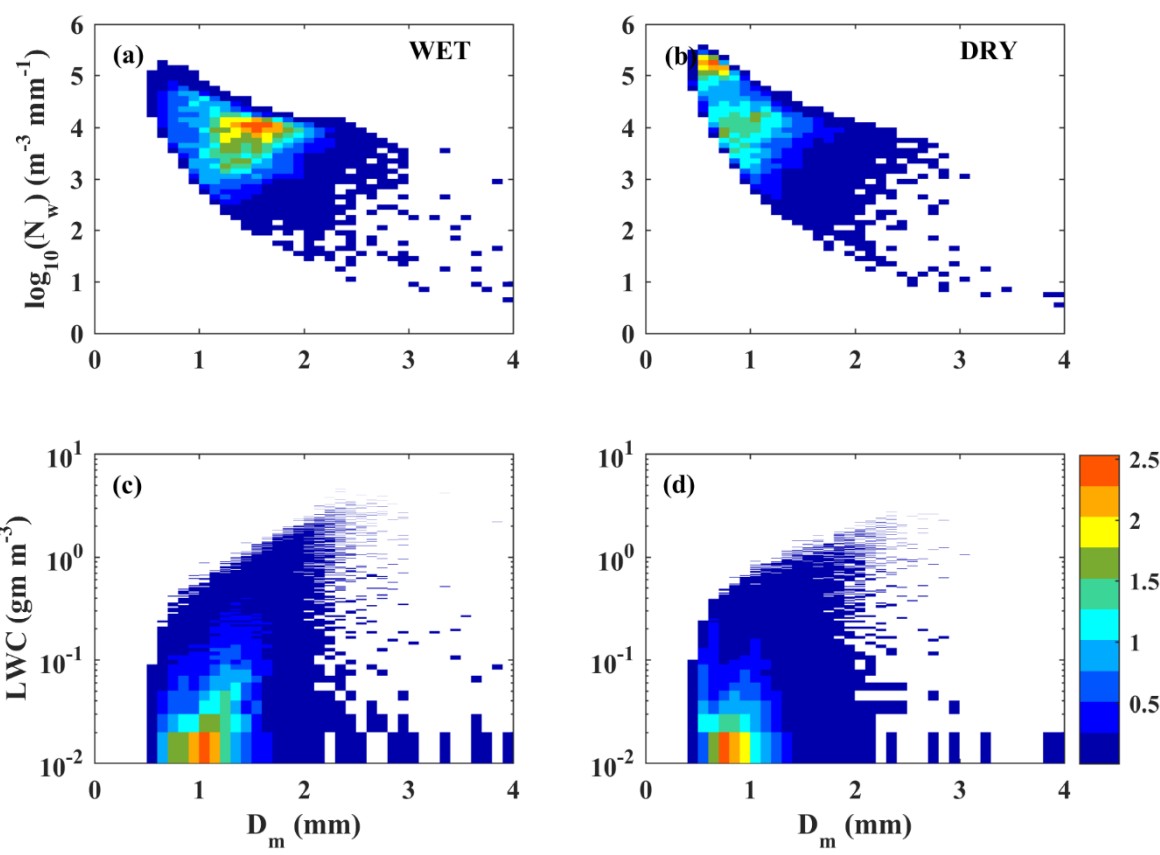


**Fig 7:** Normalized frequency of occurrence for (a)-(b) $D_m$-$\log_{10}(N_w)$ and (c)-(d) $D_m$-LWC.





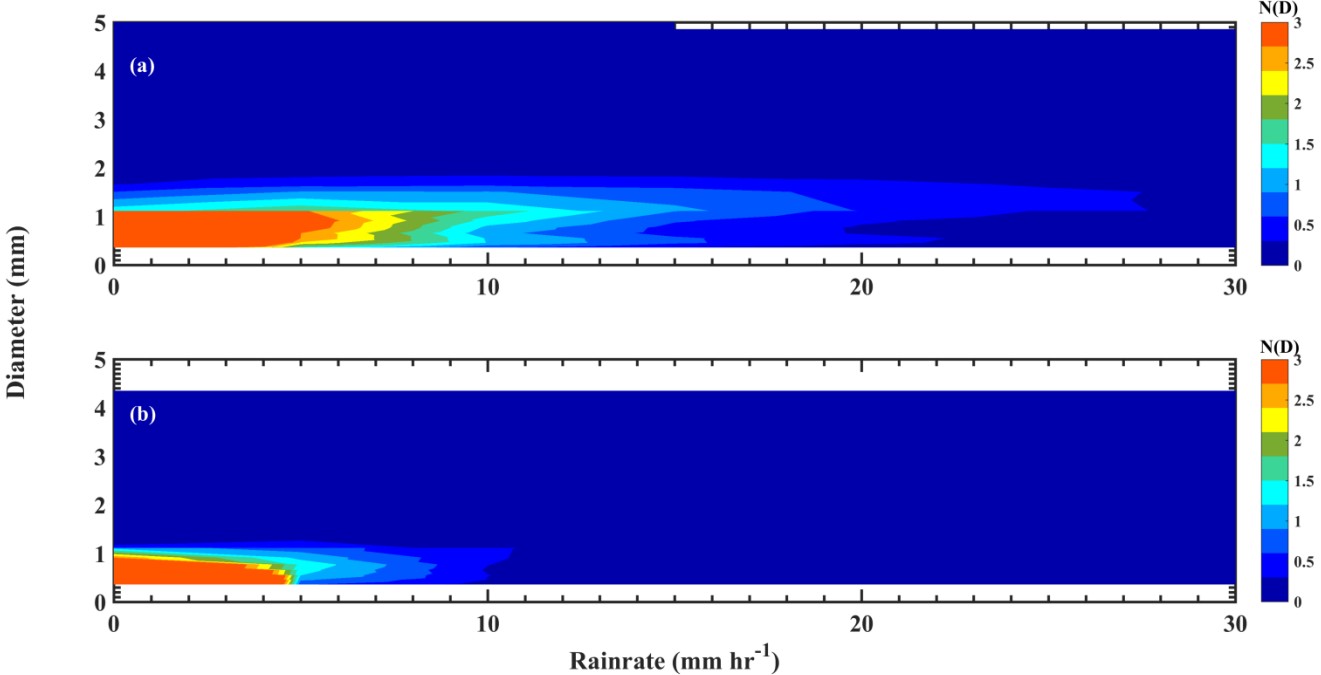

**Fig 8:** The variation in *N(D)* as a function of *D* at different *R* for (a) wet and (b) dry spells.





**Fig 9:** Normalized frequency of occurrence for (a)-(b) $D_m$-$R$ and (c)-(d) $\log_{10}(N_w)$-$R$.





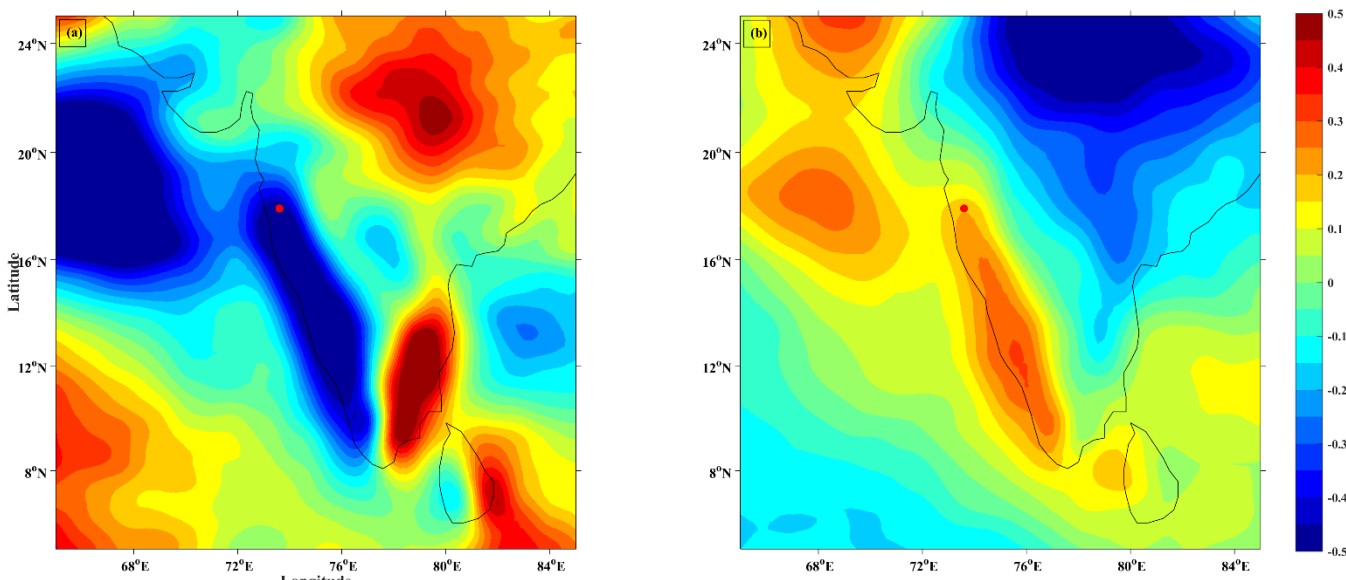

**Fig 10:** Spatial distribution of temperature anomalies (K) at 700 hPa during (a) wet and (b) dry spells of the monsoon seasons of 2012-2015. Here positive anomaly represents heating, and negative anomaly represents cooling. The red dot represents the observational site.






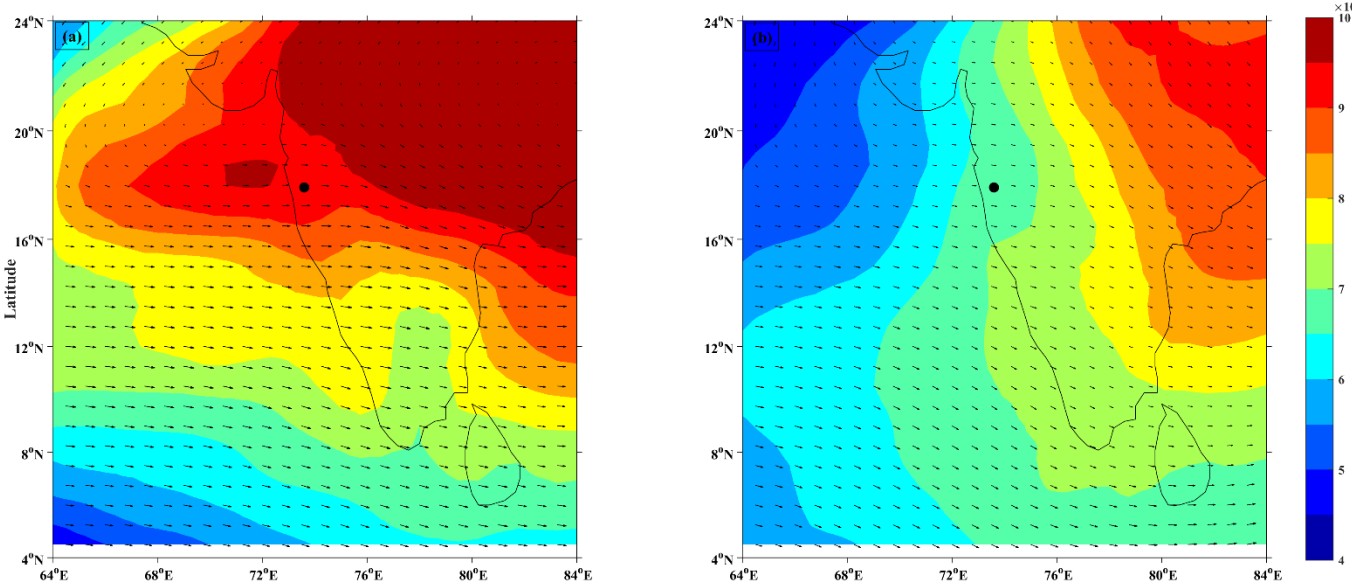

**Fig 11:** Spatial distribution of mean horizontal winds (m s$^{-1}$) and mean specific humidity (kg kg$^{-1}$) at 700 hPa during (a) wet and (b) dry spells of the monsoon seasons of 2012-2015. Here color bar represents the specific humidity, and arrows represent horizontal wind velocity. The black dot represents the observational site.





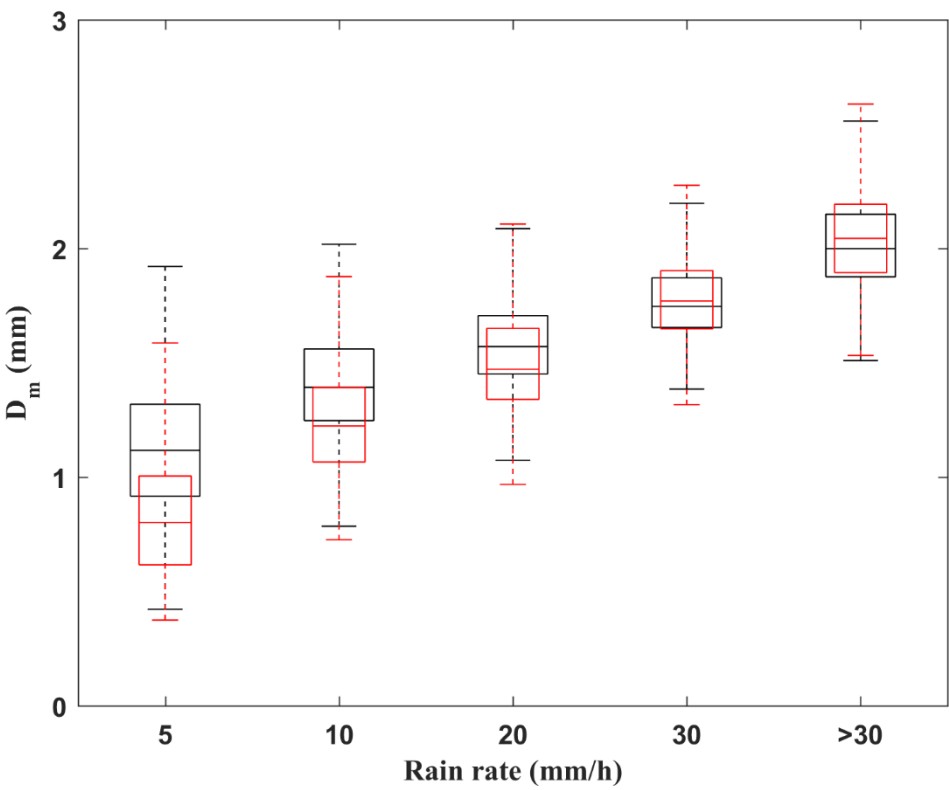

**Fig 12:** Distribution of $D_m$ at different rain rates during wet and dry spells. In this plot, the horizontal line within the box represents the median value. The boxes represent data between first and third

quartiles, and the whiskers show data from 12.5 to 87.5 percentiles. Here black color represents wet spells, and the red color represents dry spells.



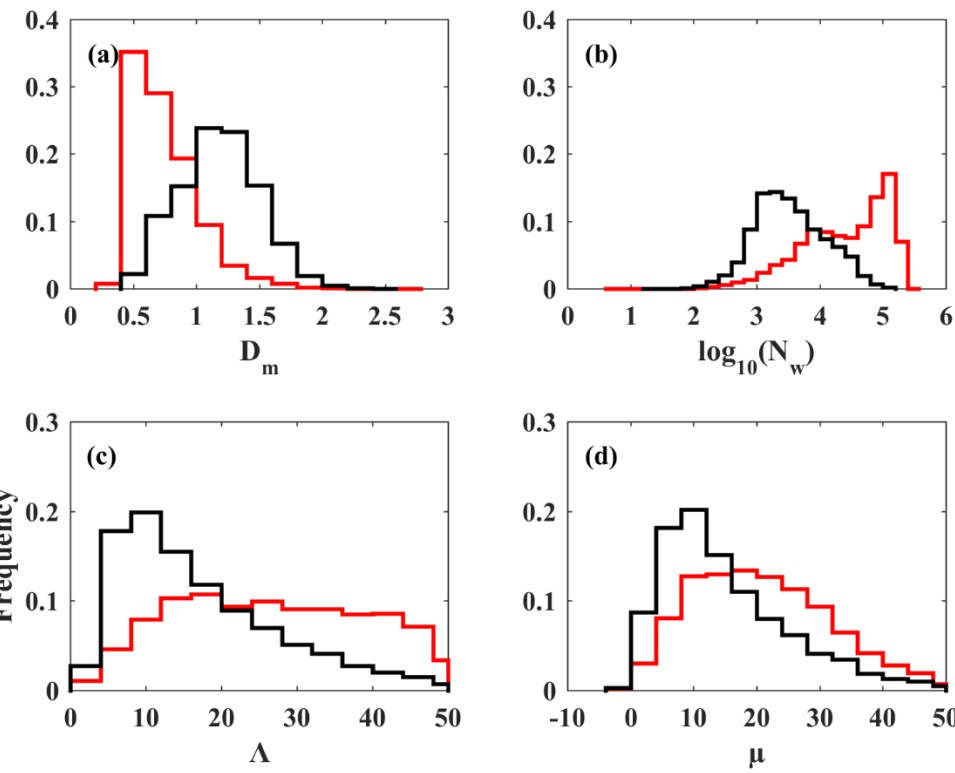

**Fig 13:** Histograms of $D_m$, $\log_{10}(N_w)$, $\Lambda$ and $\mu$ in stratiform rain during wet and dry spells. Blackline represents wet spells, and the red line represents dry spells.





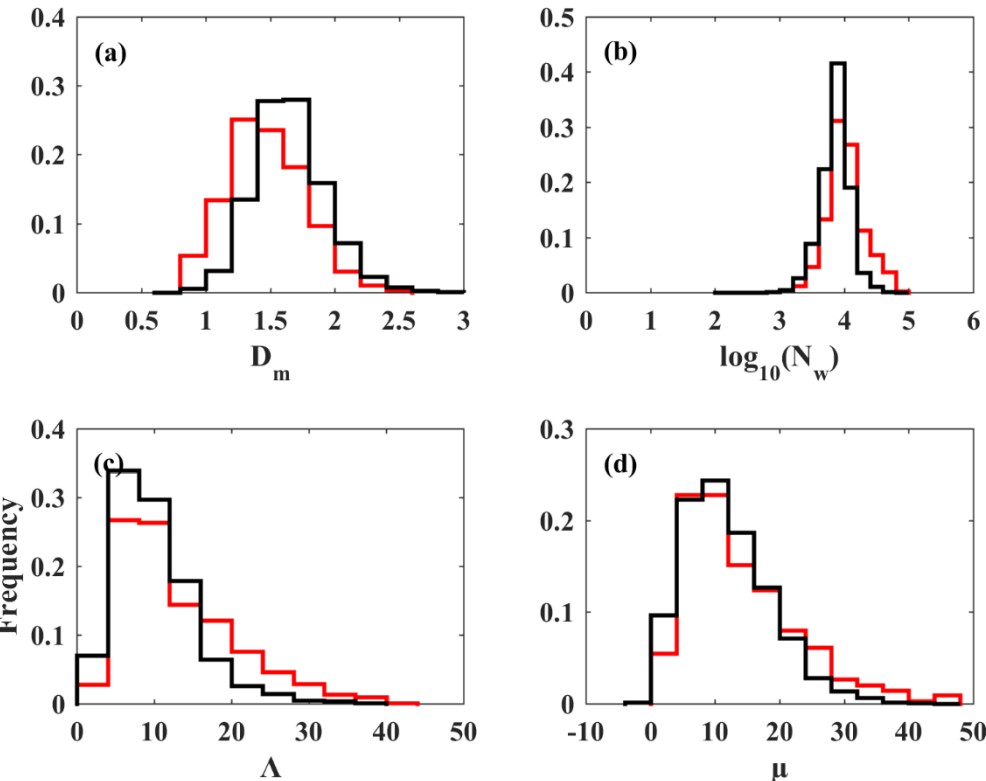

**Fig 14:** Histograms of $D_m$, $\log_{10}(N_w)$, $\Lambda$ and $\mu$ in convective rain during wet and dry spells. Blackline

1005        represents wet spells, and the red line represents dry spells.





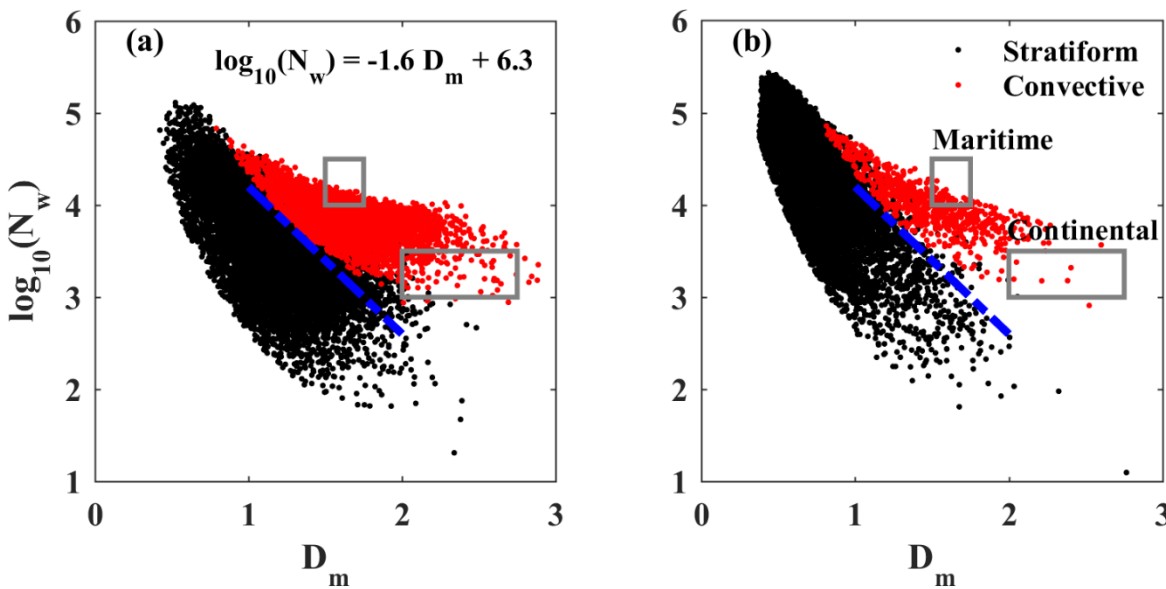

**Fig 15:** Scatterplot of $D_m$ versus $\log_{10}(N_w)$ for convective (red dots) and stratiform (black dots) rain types in (a) wet spells and (b) dry spells. The dashed line represents the stratiform line, and two gray rectangles represent maritime and continental clusters, as reported in Bringi et al. (2003). The equation separating the stratiform and convective cluster used by Bringi et al. (2003) is also provided in the figure.



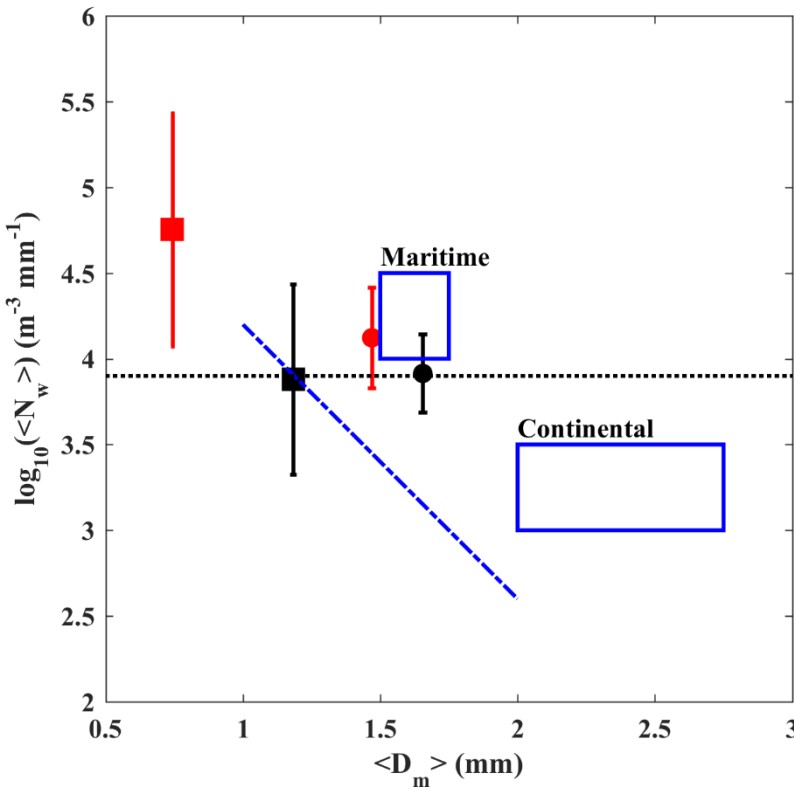

**Fig 16:** The average value of $\log_{10}(N_w)$ and $D_m$ along with standard deviations during stratiform and convective rain in wet and dry spells. The dashed line represents the stratiform line, and the rectangle boxes represent maritime and continental clusters, according to *Bringi et al.* (2003). The dotted horizontal line represents the Marshall-Palmer value of $\log_{10}(N_w)$. The black (red) color represents the wet (dry) spells. The filled circles (squares) represent the convective (stratiform) rain types.






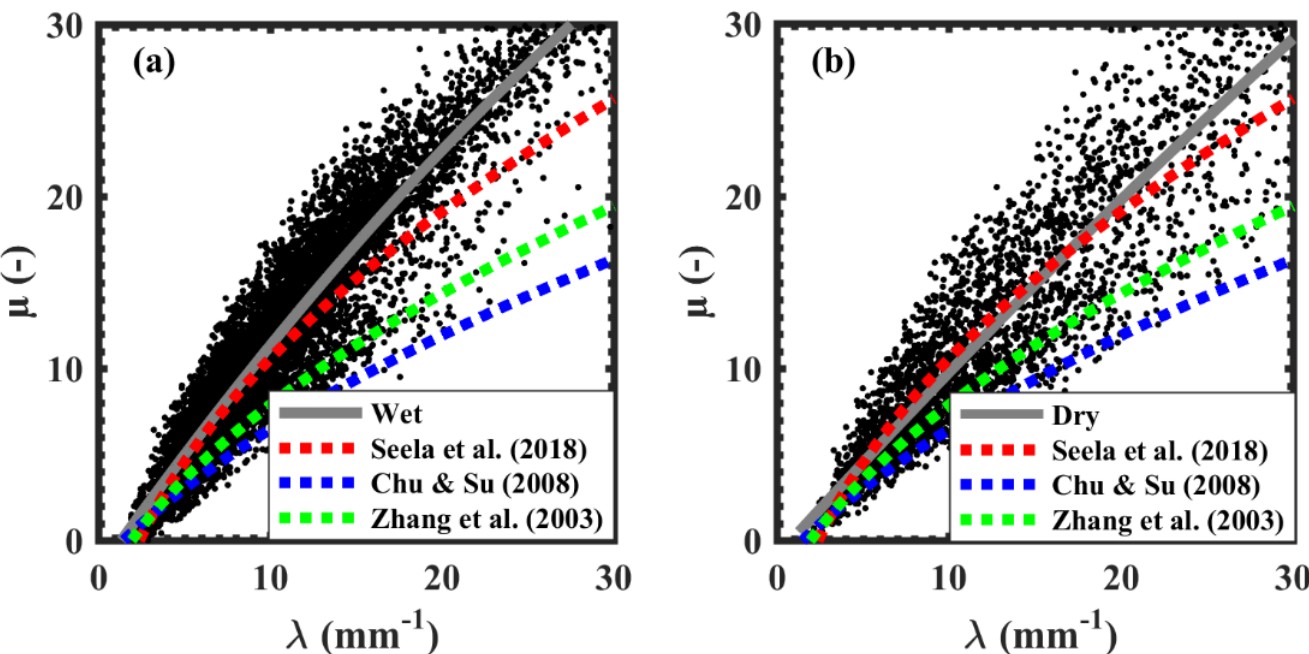


**Fig 17:** Scatter plots of $\Lambda$-$\mu$ values obtained from gamma DSD for (a) wet and (b) dry spells. The solid

line indicates the least square polynomial fit for $\Lambda$-$\mu$ relation.





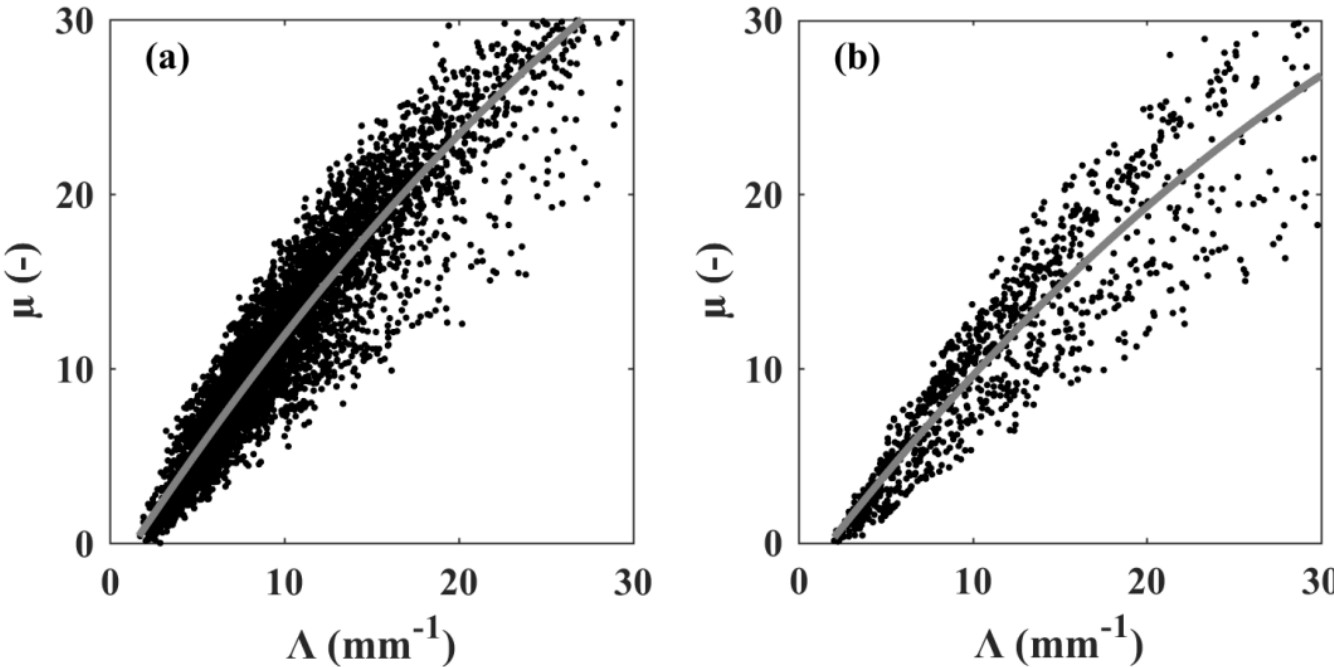

**Fig 18:** Scatter plots of $\Lambda$-$\mu$ values obtained from gamma DSD for (a) convective and (b) stratiform

rain. The solid line indicates the least square polynomial fit for $\Lambda$-$\mu$ relation.



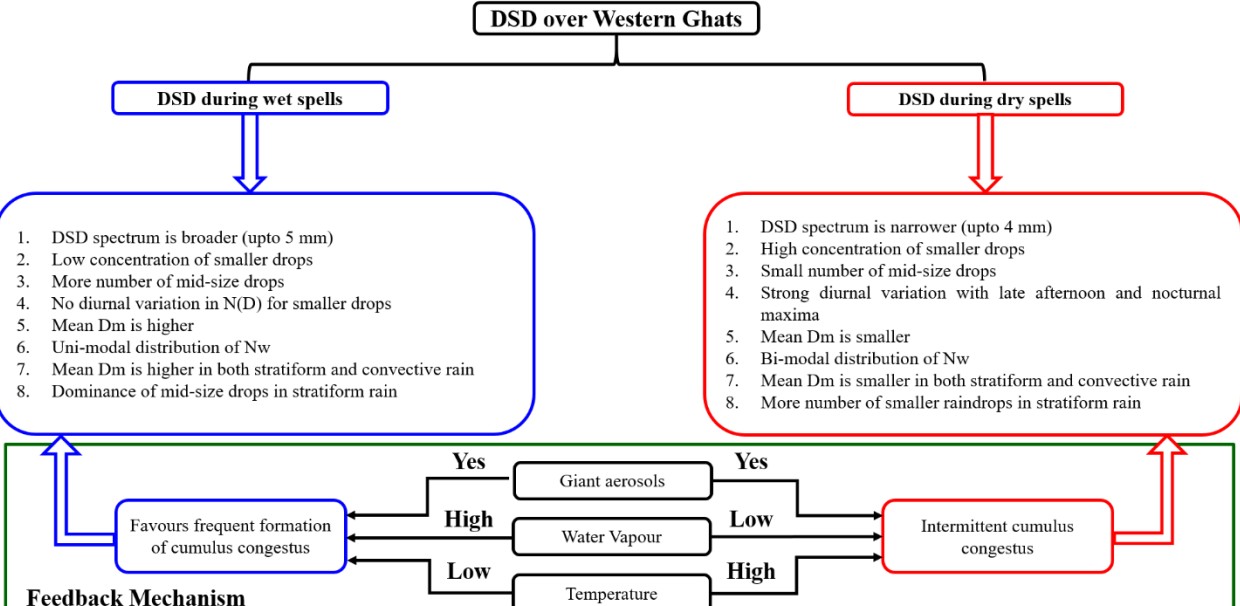

**Fig 19:** Summary of the DSD characteristics during wet and dry spells over WGs.