# Peer review of "Statistical characteristics of raindrop size distribution over Western Ghats of India: wet versus dry spells of Indian Summer Monsoon"

_Atmospheric Chemistry and Physics, 2019_

## Referee Comment (RC1) · Anonymous Referee #1 · 14 Jan 2020

Major comments: This study presents an overall analysis of the rain droplet size distribution (DSD) characteristics over the Western Ghats (WG), India. The authors contrast measurements during the relatively dry and wet periods, as well as between stratiform and convective clouds. Additionally, they propose new $\mu$-$\Lambda$ relations for the region within a context provided by previous studies. Overall, the results presented carry significance since the DSD characteristics throughout different climatic/cloud type, regimes are determinant for accurate cloud-resolving modelling. However, the manuscript requires further corrections, less Figures and better discussions. Here I provide the major issues followed by specific comments in the next section. In Section 4.1 the authors provide an analysis of the orography effect on the DSD measurements

using GPM. While I think this is an interesting study, it is not clear what motivated it and what were the conclusions. The authors showed that Dm presents a higher variability on the leeward side, which was due to the presence of deeper clouds. The deep convection in the leeside is well known and published in several papers. But it is not clear if the authors claim this is due to the orography effect, due to larger rain rate during wet spells or if there is something else. Additionally, the GPM microphysical retrievals have large errors and the readers has no information if the errors are larger than the differences or not. In addition, it is not mentioned how the authors expect the orography to affect the overall conclusions of the manuscript. For sure it is important to discuss the effect of the topography because the data was collected at the leeside, but the results presented are incomplete and has no connection with the conclusions. Going through Sections 4.2-4.5, I noticed a significant amount of repetition in the analysis. While the authors are looking at different things in those sections, the overall conclusions are fairly similar – that rain droplets tend to be smaller in the dry spells as compared to the wet spells (which is not surprising). This is repeated in each section as though they exist in isolation. There does not seem to be a logical thread motivating the reader to go through the figures and sections in order. This is representative of an issue present in the whole manuscript, which is the lack of synthetization. There is a lot of text dedicated to describing the figures, while only a small portion is dedicated to discussing the physical reasoning. I believe the authors could do a better job of describing what they are looking for in their dataset and possibly even compile Sections 4.2-4.5 into a single section. However, the main problem in this Sections are related to the bias in the results due to the different rain rate from dry and wet spells. In the last part of the manuscript the authors do a nice job, organizing the results by rain rate and cloud type. In the last part you can conclude some important issues, however, in this Sections we only see the signal of different rainfall. I suggest the authors to cut this part and add a figure showing the rain rate as function of the diurnal cycle, dry/wet spells, cloud type. These will be much more important. Another major issue regards the authors understanding of the microphysical processes explaining the DSD character-

istics. The most serious problem is a small comment starting in line 261. The authors claim that condensational growth can partly explain the presence of small raindrops, which is wrong based on the accepted theory of droplet growth and past observations. The JWD measurements provided in the manuscript deals with droplets larger than 0.3 mm, which is a size range where the condensation is negligible. While this was only a small comment, it highlighted some misconceptions about the authors understanding of droplet growth. Granted, it is very difficult to discuss microphysical processes looking only at ground-based measurements since the DSDs are the result of multiple competing mechanisms. The ideal scenario would be to provide a more interesting analysis about the measurements themselves instead of only inferring microphysical processes from the DSD measurements. This is the reason I think Section 4.6 is the strongest part of the manuscript, which introduces a contextualized analysis of the DSD parameters through the $\mu$-$\Lambda$ relations. Another important point is the no evaluation of the CCN in each situation, the authors remember in the middle of the manuscript to talk about aerosol, however, the discussion and results should take into account the aerosol effect in the DSD. In addition, the methodology to adjust a Gamma function each one minute add noise to the results, because many times one-minute distribution is not significant to describe a Gamma function. The authors should at least evaluate if an adjust each 5 minutes presents the same result. I had a lot of problem when I used one minute to adjust a Gamma function. The section describing the synoptic features for dry and wet spells is out of the scope of the paper and several others papers (cited in the manuscript) described the main synoptic differences between dry and wet spells. In addition, most of the characteristics seems to be a consequence of dry/wet conditions, and not a synoptic difference responsible for a dry/wet period, for instance, temperature is naturally higher when you have less rainfall. Furthermore, as there are differences among leeside and windward and the resolution is 0.5 degrees it can mix different situations. The conclusions of the manuscript are rather descriptive as well. It is not very clear what were the gains in understanding from this study. However, given the limited setup available, I don't think the authors will be able to make substan-

tial claims about microphysical processes. I'd rather see a bigger focus on the DSD measurements themselves, as done in Section 4.6. Based on the comments above, I suggest that the authors do major changes to the manuscript before submitting the revised version. The last Sections of the manuscript are the most important the authors can address the same contents using much less figures and be more clear, objective and precise in the study.

2- Specific comments:

L.23-24: "comparison with the earlier studies" -> "Comparison with previous studies". L.29: the first sentence is awkwardly phrased. Please review – suggestion: "The Western Ghats (WG) is one the two Indian regions subject to heavy rainfall". What is the other region? Should mention as well. L.42: What is "good rainfall"? Could mean multiple things for different readers. Change to "heavy rainfall" or something similar. L.58: "DSD is related to" -> "DSD is related to processes such as". There are other processes affecting the DSD formation (advection, ice melting, droplet breakup to name a few). L.73: How did the authors obtained geopotential height anomalies and OLR using a X-band radar? Should mention that the authors used more than X-band measurements. L.76-77: Use either "a positive geopotential height anomaly" or "positive geopotential height anomalies". Many of these features are consequence and not the driver. L.80: "speculate" -> "infer". L.84-85: What did the Konwar et al. (2014) study concluded? You only mention the shortcoming of the study and not how it motivated your manuscript. Would be nice to provide a few more details on the Harikumar (2016), Das et al. (2017) and Sumesh et al. (2019) studies as well. After summarizing their results, you can point to the knowledge gaps still remaining. L.95-97: do not have to repeat that the microphysical processes can be inferred from the DSDs. L.101: "terrain, WGs" -> "terrain of WGs". L.116: indicate the JWD was placed at the leeside. L.118: "a size" -> "sizes". L.121: on size or concentration? Or both? L.128: "the raindrops" -> "raindrops". L.128-130: how do you know there are no droplet over 5.5 mm if you can't measure them? I suggest removing the last three sentences of this paragraph. L.137: I believe the authors estimate R and Z from the JWD measurements and not "JWD estimates rain rate (R) and reflectivity (Z)...". L.151: "momentum" -> "moment". L.166: "lacking to study" -> "difficulties of studying". L.172: is the 1-degree resolution enough to solve the topography differences? L.187: is the data collected at the leeside? L.191: "the comparison" -> "comparison". L.191-193: avoid repeating "comparison". L.206: "time series are" -> "time series is". L.216: The use of model to define rainfall has some issues, because model does not describe very well the rainfall. Why you did not employed the raingauge employed in this study? L.218: "that dry days are more" -> "that there are more dry days". L.228: "windward side and leeward side" -> "windward and leeward sides". L.229: should note that you use GPM for this analysis – in the beginning of the section. L.234-237: the logic here is weird (at least the way it is written). First you mention that GPM underestimates (overestimates) Dm (Nw) and, because of that, GPM can be used for your study. Should be something like "while GPM have X and Y shortcomings (as pointed out by previous studies), it can be used for our objectives with the following caveats". L.251: what is the conclusion from this section? L.256: How the data showed in Figure 4 was normalized? It is not clear, have you used the total amount for each class of rainfall? Also, this Figure is missing large droplets. Why the authors do not show the rain rate each hour? L.258: The large number...it is normalized so I have no idea bout the total number. Fig. 4: put label in the y-axis in panel a). L.261-264: I don't agree that condensation has any effect on the JWD observations. The lowest size bin of the JWD is 0.3 mm, while condensation is mostly effective below 30-40 $\mu$m. Therefore, the "condensational growth of raindrops" is most certainly negligible – unless you provide calculations supporting your claim. L.263: This is a place to discuss the effect of the aerosol. L.265-266: evaporation can be more efficient during the dry spells but is not limited to it. L.271: If evaporation is the reason, why it is similar every time, during the cloud formation it should not be as important as during dissipation phase. L.274: could there be ice melting as well? L.267-268: I don't think negative $\Lambda$ values were ever reported in the literature. L.299: this sentence is missing something. It has no relation with the former discussion L.302: At this point

of the study you have not presented any cloud type discussion and this result could be due to different cloud type. L.308: your scale goes up to 3.0 mm!!!! how can you talk about 4.8 mm? L. 320: The wet spells also have a bimodal feature, actually this is a typical feaure of tropical convection. L.342: I would argue that the shallow convection is mostly due to the weak updrafts and not the collision-coalescence process. L.480: The results now depends from rain rate as already explained. The authors should re-organized the manuscript in order to have an easy flow of useful information. L.520: "remote" -> "remote sensing". L.580: "small Dm, large Nw" -> "small Dm and large Nw".

---

## Referee Comment (RC2) · Anonymous Referee #2 · 5 Mar 2020

Title: Statistical characteristics of raindrop size distribution over Western Ghats of India: wet versus dry spells of Indian Summer Monsoon

Overall Comments: This manuscript presents a detailed analysis of the raindrop size distribution (DSD) during wet and dry spells of the Indian Summer Monsoon over Western Ghats of India. The DSD data are collected from a Joss-Waldvogel disdrometer during June-September of 2012-2015. DSD characteristics, including the diurnal variation and DSD spectra at different rain rates, as well as the gamma distribution parameters in different types of rainfall (i.e., stratiform vs convective) are summarized.

Overall, this paper is easy to follow. However, it is still not clear what the authors have

addressed in addition to showing the DSD characteristics. Further discussions are required to elaborate the physical reasoning in this analysis.

Specific Comments:

First of all, I do not really see anything new in this manuscript. The analysis methods are rather conventional, and the findings are not well highlighted. The authors should elaborate more on the motivation and novelty of this study. More importantly, the authors should provide sufficient discussion and explanation about the DSD characteristics indicated in the observations.

In addition, I have serous concerns about using single point data to resolve the orographic enhancement. Justification is required from this perspective.

I am not convinced by the interpretation of the orographic gradients using GPM products. Also, the authors should incorporate the uncertainties in GPM retrievals. In fact, I do not really think this manuscript will have significant impact without including more in-situ data.

The authors have included analyses and figures from many perspectives. But, none of them has been detailed in a very quantitative manner. In fact, the reviewer was not even clear about what problems the authors are trying to address except the detailed illustration of DSD characteristics. Significant revisions are requested to highlight the scientific merits.

Anyway, the authors started from an interesting point about orographic precipitation. But I did not see sufficient investigation on this aspect.

---

## Author Comment (AC1) · 28 Mar 2020

**Response to Reviewer # 1s' Comments**

| Comm./ Response | Descriptions |
|---|---|

**Major Comments**

**General Com.** *This study presents an overall analysis of the rain droplet size distribution (DSD) characteristics over the Western Ghats (WG), India. The authors contrast measurements during the relatively dry and wet periods, as well as between stratiform and convective clouds. Additionally, they propose new μ-Λ relations for the region within a context provided by previous studies. Overall, the results presented carry significance since the DSD characteristics throughout different climatic/cloud type, regimes are determinant for accurate cloud-resolving modelling. However, the manuscript requires further corrections, less Figures and better discussions. Here I provide the major issues followed by specific comments in the next section.*

**Response** **We would like to sincerely thank the anonymous referee for his critical examination and constructive comments/suggestions on the manuscript. We went through all the referee comments and suggestions and implemented the same in the revised manuscript. Point-to-point clarifications for the referee's comments and how we have addressed each recommendation are given below. The manuscript is also edited by considering the other referee's comments.**

**Comm. # 1a** *In Section 4.1 the authors provide an analysis of the orography effect on the DSD measurements using GPM. While I think this is an interesting study, it is not clear what motivated it and what were the conclusions.*

**Response** **This manuscript aims to examine the characteristics of raindrop size distributions (DSDs) using disdrometer measurements during the drier and wetter periods of the Indian Summer Monsoon (ISM) on the Western Ghats (WGs) region. Earlier studies have shown that the complex topography can modulate the WGs precipitation processes (e.g., Konwar et al., 2014; Utsav et al., 2017, etc.). The GPM analysis is used to provide an overview of the DSD over this topographic region irrespective of wet and dry spells of ISM. Such a review is not possible with single point-wise observation. Examining the orographic impact on DSDs was one of the reviewer requirements, which was addressed and included in the manuscript. Authors also believe that such analysis is useful in understanding the underlying rain microphysical processes (indirect inference on collision-coalescence, breakup, evaporation, etc. that can shape the DSD) at different altitudes. Moreover, the parameterization schemes in the numerical models also demand the dominant microphysical processes like collision-coalescence, breakup, etc. at a finer resolution. Hence, it is necessary to have the rain microphysical observations at different locations over the complex topography.**

**We restrict the GPM analysis over three areas viz., ocean, windward, and leeward side of the mountain. It is found that the DSD distribution is smaller over the ocean and windward side, whereas the DSD shows large variability on the leeward side. The results are consistent with the ground-based measurement of DSDs over**

similar topography observed during the Olympic Mountains Experiment (OLYMPEX) (Zagrodnik et al., 2019).
**The relative sentences have been added in the revised manuscript.**

**Comm. # 1b** *The authors showed that Dm presents a higher variability on the leeward side, which was due to the presence of deeper clouds. The deep convection in the leeside is well known and published in several papers. But it is not clear if the authors claim this is due to the orography effect, due to larger rain rate during wet spells or if there is something else.*

**Response** **In this work, from GPM measurements, we presented the background DSD characteristics over the complex topography for the monsoon season, not for the active and break spells of ISM. Limited data from GPM measurement restrict our analysis to examine the role and impact of orography on the DSD distribution during wet and dry periods of ISM. At this point, the reviewer's suggestion can be considered for future work.**

**Comm. # 1c** *Additionally, the GPM microphysical retrievals have large errors and the readers has no information if the errors are larger than the differences or not.*

**Response** **The GPM-DPR can estimate the rain integral parameters, $D_m$, and $N_w$ using dual-frequency ratio (DFR) method. However, the GPM-DPR suffers shortcomings/limitations. The DSD parameterization used in the GPM-DPR is the gamma distribution with a constant shape parameter, μ=3. The constant value of 'μ' introduces errors in the retrievals. The retrieval of $D_m$ using DFR method is iterative and the $D_m$ has two solutions when the DFR is less than 0. The uncertainties in the GPM-DPR in estimating the DSD are detailed in Seto et al. (2013), Liao et al. (2014), etc. Krishna et al. (2017) assessed the DSD measurements from GPM over WGs by comparing with the ground-based disdrometer. They showed that the seasonal variations in $D_m$ and $N_w$ are well represented in the GPM measurements; however, underestimate in comparison to the ground-based disdrometer measurement. Radhakrishna et al. (2016) also found that the GPM underestimates and overestimates the mean $D_m$ and N$w$ during the southwest and northeast monsoons in Gadanki, a semiarid region of India. They showed that the single-frequency algorithm underestimates the mean $D_m$ by ~0.1 mm below 8 mm hr$^{-1}$, and the underestimation is little higher at higher rain rates. Whereas in the dual-frequency algorithm, the mean $D_m$ is nearly same below 8 mm hr$^{-1}$, but underestimates (~0.1 mm) at higher rain rates. Further, the underestimation is very small for $D_m$ values below 1.5 mm. In the present study, most of the $D_m$ values present below 1.5 mm. Here, the authors intend to show the variability of DSDs over the complex terrain, not the absolute value. Hence, it is reasonable to consider the GPM measurements to present the overview of DSD over the complex topography.**

**The relative sentences have been added in the revised manuscript.**

**Comm. # 1d** *In addition, it is not mentioned how the authors expect the orography to affect the overall conclusions of the manuscript. For sure it is important to discuss the effect of the topography because the data was collected at the leeside, but the results presented are incomplete and has no connection with the conclusions.*

**Response** **The single point-wise instrument deployed at the mountain peak of WGs is not sufficient to address the orographic impacts on the DSDs. One of the difficulties in studying the effect of orography is the unavailability of the in-situ data (like disdrometer network) on the WGs region (windward and leeward slopes). It is a well-accepted suggestion to examine the effect of orography on DSDs by deploying the observational network of disdrometers and rain gauges along the windward and leeward slopes of the WGs. This suggestion can be addressed in future works.**

**Comm. # 2** *Going through Sections 4.2-4.5, I noticed a significant amount of repetition in the analysis. While the authors are looking at different things in those sections, the overall conclusions are fairly similar – that rain droplets tend to be smaller in the dry spells as compared to the wet spells (which is not surprising). This is repeated in each section as though they exist in isolation. There does not seem to be a logical thread motivating the reader to go through the figures and sections in order. This is representative of an issue present in the whole manuscript, which is the lack of synthetization. There is a lot of text dedicated to describing the figures, while only a small portion is dedicated to discussing the physical reasoning. I believe the authors could do a better job of describing what they are looking for in their dataset and possibly even compile Sections 4.2-4.5 into a single section.*

**Response** **Thanks for the suggestion.**
**The manuscript is edited thoroughly, and sincere efforts have been taken to avoid the repetitions. The manuscript is re-organized into different sections by synthesizing the figures. The results are discussed in detail for the figures.**
**In the revised manuscript, sections 4.2-4.5 are compiled into two sections.**

**Comm. # 3** *However, the main problem in this Sections are related to the bias in the results due to the different rain rate from dry and wet spells. In the last part of the manuscript the authors do a nice job, organizing the results by rain rate and cloud type. In the last part you can conclude some important issues, however, in this Sections we only see the signal of different rainfall. I suggest the authors to cut this part and add a figure showing the rain rate as function of the diurnal cycle, dry/wet spells, cloud type. These will be much more important.*

**Response** **The authors appreciate the suggestion.**
**The manuscript is re-organized to have a smooth flow with the text and figures. The necessary discussion for the results is provided, and the diurnal variation of rain rate during wet and dry spells is also included.**
**The relative sentence and figure have been added in the revised manuscript.**

**Comm. # 4** *Another major issue regards the authors understanding of the microphysical processes explaining the DSD characteristics. The most serious problem is a small comment starting in line 261. The authors claim that condensational growth can partly explain the presence of small raindrops, which is wrong based on the accepted theory of*

*droplet growth and past observations. The JWD measurements provided in the manuscript deals with droplets larger than 0.3 mm, which is a size range where the condensation is negligible. While this was only a small comment, it highlighted some misconceptions about the authors understanding of droplet growth.*

**Response**      **We agree with the reviewers that the condensational growth is negligible at 0.3 mm diameter. However, in line 261, the authors refer to the background microphysical processes during the formation of raindrops within the cloud environment.**

**As per the referee's comment, "... condensational growth can partly explains the presence of small drops" is deleted in the revised manuscript as it is misleading the readers.**

**Comm. # 5**      *Granted, it is very difficult to discuss microphysical processes looking only at ground-based measurements since the DSDs are the result of multiple competing mechanisms. The ideal scenario would be to provide a more interesting analysis about the measurements themselves instead of only inferring microphysical processes from the DSD measurements. This is the reason I think Section 4.6 is the strongest part of the manuscript, which introduces a contextualized analysis of the DSD parameters through the μ-Λ relations. Another important point is the no evaluation of the CCN in each situation, the authors remember in the middle of the manuscript to talk about aerosol, however, the discussion and results should take into account the aerosol effect in the DSD.*

**Response**      **Thanks for the suggestion.**

**As there is no direct evidence for the aerosols and CCN for this study, the text related to aerosols and CCN is deleted from the manuscript.**

**Comm. # 6**      *In addition, the methodology to adjust a Gamma function each one minute add noise to the results, because many times one-minute distribution is not significant to describe a Gamma function. The authors should at least evaluate if an adjust each 5 minutes presents the same result. I had a lot of problem when I used one minute to adjust a Gamma function.*

**Response**      **Thanks for the suggestion.**

**The DSD measurements from JWD are adjusted to 5-min and fitted with the gamma distribution. The histogram of $D_m$ for 1-min and 5-min gamma distribution for the study period is shown in Figure 1. It is clear that the 1-min and 5-min gamma DSD shows a similar distribution, and significant differences do not exist between them. If the data is divided into wet and dry spells, the distribution could be the same, and hence the 1-min gamma DSD can be used in the present study without hesitation.**

[Figure]

*Fig. 1: Normalized distribution of $D_m$ obtained from 1-min and 5-min gamma distribution function.*

| **Comm. # 7** | *The section describing the synoptic features for dry and wet spells is out of the scope of the paper and several others papers (cited in the manuscript) described the main synoptic differences between dry and wet spells. In addition, most of the characteristics seems to be a consequence of dry/wet conditions, and not a synoptic difference responsible for a dry/wet period, for instance, temperature is naturally higher when you have less rainfall.* |
|---|---|
| **Response** | **We agree with the referee that the dynamical features are consequences of wet/dry monsoon spell. However, these dynamical features play an important role in modifying the DSD properties during the wet/dry periods. Hence, these synoptic differences are provided in the manuscript to better understand the DSD characteristics during the wet and dry spells.** |
| **Comm. # 8** | *Furthermore, as there are differences among leeside and windward and the resolution is 0.5 degrees it can mix different situations. The conclusions of the manuscript are rather descriptive as well. It is not very clear what were the gains in understanding from this study. However, given the limited setup available, I don't think the authors will be able to make substantial claims about microphysical processes. I'd rather see a bigger focus on the DSD measurements themselves, as done in Section 4.6. Based on the comments above, I suggest that the authors do major changes to the manuscript before submitting the revised version. The last Sections of the manuscript are the most important the authors can address the same contents using much less figures and be more clear, objective and precise in the study.* |

| | |
|---|---|
| **Response** | **Thanks for the suggestion.** |
| | **The authors took at most care while considering the windward and leeward sides in analyzing the DSD characteristics. So the results can be viewed as a representative of DSD characteristics on the windward and leeward sides in a broader (statistical) perspective. The major challenge in studying the orographic effect on the DSD is the unavailability of the disdrometer network deployed in the windward and leeward sides of the WGs, which could really capture the topography variations across the WGs. The rain microphysical characteristics may vary in different phases of the ISM due to different dynamical mechanisms of rain formation. Still, there are no studies that exist to demonstrate the differences in DSDs during different phases of ISM. This work is the first study that quantifies the DSD characteristics during the drier and wetter periods of the ISM in the WGs region.** |
| | **As per the reviewer's suggestion, the authors put their sincere effort to concise the manuscript with the clear objectives and better representation of results.** |

**Specific comments**

| | |
|---|---|
| **Comm. # 1** | *L.23-24: "comparison with the earlier studies" -> "Comparison with previous studies".* |
| **Response** | **The sentence is modified as "The $\Lambda$-$\mu$ relations obtained in the present study are slightly different in comparison with previous studies."** |

| | |
|---|---|
| **Comm. # 2** | *L.29: the first sentence is awkwardly phrased. Please review – suggestion: "The Western Ghats (WG) is one the two Indian regions subject to heavy rainfall". What is the other region? Should mention as well.* |
| **Response** | **The typographical error is corrected in the manuscript.** |
| | **Western Ghats (WGs) is one of the heavy rainfall regions in India.** |

| | |
|---|---|
| **Comm. # 3** | *L.42: What is "good rainfall"? Could mean multiple things for different readers. Change to "heavy rainfall" or something similar.* |
| **Response** | **The sentence is modified as "The ISM shows large spatial and temporal variability. It is well known that during the active (with a high amount of rain) and break (little or no rain) spells of the ISM."** |

| | |
|---|---|
| **Comm. # 4** | *L.58: "DSD is related to" -> "DSD is related to processes such as". There are other processes affecting the DSD formation (advection, ice melting, droplet breakup to name a few).* |
| **Response** | **The sentence is modified as "Raindrop size distribution (DSD) is a fundamental microphysical property of the precipitation. DSD is related to processes such as hydrometeor condensation, coalescence, and evaporation..."** |

| | |
|---|---|
| **Comm. # 5** | *L.73: How did the authors obtained geopotential height anomalies and OLR using a X-band radar? Should mention that the authors used more than X-band measurements.* |
| **Response** | **The reviewer's suggestion is implemented. The sentence is modified as "... X-band radar observations and other ancillary datasets."** |

**Comm. # 6**    *L.76-77: Use either "a positive geopotential height anomaly" or "positive geopotential height anomalies". Many of these features are consequence and not the driver.*

**Response**    **Referee's suggestion is implemented.**

**Comm. # 7**    *L.80: "speculate" -> "infer".*

**Response**    **Reviewer's suggestion is implemented.**

**Comm. # 8**    *L.84-85: What did the Konwar et al. (2014) study concluded? You only mention the shortcoming of the study and not how it motivated your manuscript. Would be nice to provide a few more details on the Harikumar (2016), Das et al. (2017) and Sumesh et al. (2019) studies as well. After summarizing their results, you can point to the knowledge gaps still remaining.*

**Response**    **A few attempts have been made to understand the DSD characteristics over WGs. For example, Konwar et al. (2014) studied the DSD characteristics by fitting three-parameter gamma function during monsoon season. They observed that bimodal and monomodal DSDs distribution during low and high rainfall rates. However, their study is limited to brightband and non-brightband conditions only. Harikumar (2016) studied the differences between DSD on the coastal (Kochi) and high altitude station (Munnar) station located in the WGs using the lognormal fit to DSD. He observed that for a given rain rate, more number of larger size drops are present at Munnar than at Kochi. Das et al. (2017) studied the DSD characteristics during different precipitating systems on the WGs region using Disdrometer and Micro Rain Radar measurements. They noticed different reflectivity and rainfall rate relations during different types of precipitation. Sumesh et al. (2019) studied the DSD differences between mid- (Braemore, 400 m above MSL), and high-altitude (Rajamallay, 1820 m above MSL) regions in southern WGs during brightband events. They noticed bimodal DSD in mid-altitude station and monomodal DSD in the high altitude station. This study also confined to stratiform rain only.**

**There are limited studies of DSDs exist in the WGs region by considering long-term dataset. In addition, this is the first study to analyse the DSD characteristics by considering the monsoon intra-seasonal oscillations (wet and dry spells). The present study brings out the results of a unique opportunity by analysing a more extensive data set and also considering the different phases of the monsoon intra-seasonal oscillations over the WGs.**

**The above sentences are added in the revised manuscript.**

**Comm. # 9**    *L.95-97: do not have to repeat that the microphysical processes can be inferred from the DSDs.*

**Response**    **Thanks for the suggestion. The manuscript is revised.**

**Comm. # 10**    *L.101: "terrain, WGs" -> "terrain of WGs".*

**Response**    **Reviewer's suggestion is implemented.**

**Comm. # 11**    *L.116: indicate the JWD was placed at the leeside.*

**Response**    **The JWD is installed on the windward slopes of the WGs. The necessary sentence**

**is added in the revised manuscript.**

**Comm. # 12**    *L.118: "a size" -> "sizes".*
**Response**      **Referee's suggestion is implemented.**

**Comm. # 13**    *L.121: on size or concentration? Or both?*
**Response**      **The accuracy of JWD is 5% of the measured drop diameter.**
           **The necessary sentence is revised in the manuscript.**

**Comm. # 14**    *L.128: "the raindrops" -> "raindrops".*
**Response**      **Referee's suggestion is implemented.**

**Comm. # 15**    *L.128-130: how do you know there are no droplet over 5.5 mm if you can't measure them? I suggest removing the last three sentences of this paragraph.*
**Response**      **Thanks for the suggestion. The sentences "JWD cannot detect the raindrop with diameter larger than 5.5 mm ..." are deleted in the manuscript.**

**Comm. # 16**    *L.137: I believe the authors estimate R and Z from the JWD measurements and not "JWD estimates rain rate (R) and reflectivity (Z): : :".*
**Response**      **The rain rate ($R$) and reflectivity ($Z$) are estimated from JWD measurements by assuming that the momentum is entirely due to the terminal fall velocity of the raindrops.**

**Comm. # 17**    *L.151: "momentum" -> "moment".*
**Response**      **Reviewer's suggestion is implemented.**

**Comm. # 18**    *L.166: "lacking to study" -> "difficulties of studying".*
**Response**      **The sentence is modified as "One of the difficulties in studying the effect of orography is the unavailability of the disdrometer network deployed in the windward and leeward side of the WGs, which could capture the topography variations across the WGs."**

**Comm. # 19**    *L.172: is the 1-degree resolution enough to solve the topography differences?*
**Response**      **GPM level 3 data provides $D_m$ and $N_w$ values at $0.25^o$ resolution.**

**Comm. # 20**    *L.187: is the data collected at the leeside?*
**Response**      **IMD installed rain gauge at HACPL, Mahabaleshwar, which is on the windward side of the WGs.**

**Comm. # 21**    *L.191: "the comparison" -> "comparison".*
**Response**      **Referee's suggestion is implemented.**

**Comm. # 22**    *L.191-193: avoid repeating "comparison".*
**Response**      **Referee's suggestion is implemented.**

**Comm. # 23**    *L.206: "time series are" -> "time series is".*

| | |
|---|---|
| **Response** | **Referee's suggestion is implemented.** |

| | |
|---|---|
| **Comm. # 24** | *L.216: The use of model to define rainfall has some issues, because model does not describe very well the rainfall. Why you did not employed the raingauge employed in this study?* |
| **Response** | **IMD deployed numerous rain gauges over entire India. The daily rainfall data collected by these rain gauges were gridded and are available at $0.25^o \times 0.25^o$. The present study utilized the gridded rainfall data collected by IMD rain gauges to identify the wet and dry spells of ISM.**
 **The necessary sentences are added in the revised manuscript.** |

| | |
|---|---|
| **Comm. # 25** | *L.218: "that dry days are more" -> "that there are more dry days".* |
| **Response** | **Thanks for the suggestion.**
 **The sentence is modified as "It is observed that there is more number of dry days during 2012-2015 monsoon seasons."** |

| | |
|---|---|
| **Comm. # 26** | *L.228: "windward side and leeward side" -> "windward and leeward sides".* |
| **Response** | **Referee's suggestion is implemented.** |

| | |
|---|---|
| **Comm. # 27** | *L.229: should note that you use GPM for this analysis – in the beginning of the section.* |
| **Response** | **Referee's suggestion is implemented.** |

| | |
|---|---|
| **Comm. # 28** | *L.234-237: the logic here is weird (at least the way it is written). First you mention that GPM underestimates (overestimates) Dm (Nw) and, because of that, GPM can be used for your study. Should be something like "while GPM have X and Y shortcomings (as pointed out by previous studies), it can be used for our objectives with the following caveats".* |
| **Response** | **Thanks for the comment.**
 **Krishna et al. (2017) assessed the DSD measurements from GPM over WGs by comparing them with the ground-based disdrometer. They showed that the seasonal variations in $D_m$ and $N_w$ are well represented in the GPM measurements; however, underestimate in comparison to the ground-based disdrometer measurement. With the above caveats, the GPM measurements can be used to have an overview of DSD characteristics on the windward and leeward sides of WGs in a broader (statistical) perspective. Therefore, the authors attempt to present a general overview of the DSD over three different topographic regions of WGs.**
 **The necessary sentences are added in the revised manuscript.** |

| | |
|---|---|
| **Comm. # 29** | *L.251: what is the conclusion from this section?* |
| **Response** | **We perform the GPM analysis over three areas viz., ocean, windward, and leeward side of the mountain. It is found that the DSD distribution is smaller over the ocean and windward side, whereas the DSDs show large variability over the leeward side. The results are consistent with the ground-based measurement of DSDs over similar topography observed during the Olympic Mountains Experiment (OLYMPEX) (Zagrodnik et al., 2019).** |

**Comm. # 30**   *L.256: How the data showed in Figure 4 was normalized? It is not clear, have you used the total amount for each class of rainfall? Also, this Figure is missing large droplets. Why the authors do not show the rain rate each hour?*

**Response**   **The data in each hour is normalized with the total number of data in the respective spell.**

**The concentration of larger drops (above 4 mm) is lesser compared to small (below 1 mm) and mid-size (1-4 mm) drops. So it is not statistically significant to show the larger drops compared to small and mid-size drops.**

**The diurnal variation in the mean rain rate during the wet and dry spells is shown in Figure 2. The mean rain rate is higher during the wet spells throughout the day. The relatively lower rain rates in the dry spells represent the presence of a large number of smaller drops during the dry periods. The diurnal variation in rain rate shows bi-modal distribution during both wet and dry spells. The primary maximum is in the afternoon hours and the secondary maximum present during morning hours. The higher rain rates during the morning (0200-0800 LT) and afternoon hours (1400-1800 LT) represents the higher concentration of mid-size drops during the morning and afternoon hours (1400-1800 LT) during wet and dry spells.**

[Figure]

*Fig. 2: Diurnal variation of mean rain rate (mm hr$^{-1}$) during wet and dry spells.*

**The necessary sentences are added in the revised manuscript.**

**Comm. # 31**   *L.258: The large number: : :it is normalized so I have no idea bout the total number. Fig. 4: put label in the y-axis in panel a).*

**Response**   **A total of 113483 (104292) number of smaller drops (below 1 mm) present during dry (wet) spells.**
**The sentence is revised in the manuscript. The higher concentration of small drops in dry spells represents the orographic convection over WGs.**

**Comm. # 32**   *L.261-264: I don't agree that condensation has any effect on the JWD observations. The lowest size bin of the JWD is 0.3 mm, while condensation is mostly effective below 30-40 μm. Therefore, the "condensational growth of raindrops" is most certainly negligible – unless you provide calculations supporting your claim.*

**Response**   **We agree with the referee's comment that the condensational growth is negligible at 0.3 mm diameter. However, in line 261, the authors refer to the background microphysical processes during the formation of raindrops within the cloud environment.**
**As per the referee's comment, "... condensational growth can partly explains the presence of small drops" is deleted in the revised manuscript as it can mislead the results.**

**Comm. # 33**   *L.263: This is a place to discuss the effect of the aerosol.*

**Response**   **As there is no direct evidence for the aerosol and CCN during the study, the text related to CCN is deleted from the manuscript.**

**Comm. # 34**   *L.265-266: evaporation can be more efficient during the dry spells but is not limited to it.*

**Response**   **The sentence is revised as "... the large number of small raindrops during dry spells indicates that the breakup and evaporation processes may be more efficient during dry spells."**

**Comm. # 35**   *L.271: If evaporation is the reason, why it is similar every time, during the cloud formation it should not be as important as during dissipation phase.*

**Response**   **The smaller drops are omnipresent during the wet spells. The presence of smaller drops throughout the day could be due to the frequent occurrence of shallow clouds over WGs.**

**Comm. # 36**   *L.274: could there be ice melting as well?*

**Response**   **Yes, there could be ice melting, however, the chances are less as shallow clouds are dominating during the wet spell.**

**Comm. # 37**   *L.267-268: I don't think negative Λ values were ever reported in the literature.*

**Response**   **Thanks for the suggestion.**
**The negative Λ values were not shown in the figure.**

**Comm. # 38**   *L.299: this sentence is missing something. It has no relation with the former discussion*

**Response**   **Thanks for the suggestion. The sentence is deleted in the revised manuscript.**

**Comm. # 39** *L.302: At this point of the study you have not presented any cloud type discussion and this result could be due to different cloud type.*

**Response** **Thanks for the suggestion. The sentence is deleted in the revised manuscript.**

**Comm. # 40** *L.308: your scale goes up to 3.0 mm!!!! how can you talk about 4.8 mm?*

**Response** **The $D_m$ value varies upto 4.8 mm. However, their presence is very small above 3 mm. Hence, the distribution is restricted upto 3 mm in the figure.**

**Comm. # 41** *L. 320: The wet spells also have a bimodal feature, actually this is a typical feature of tropical convection.*

**Response** **Yes, there is a bimodal distribution in wet spells, however, their amplitude is very small.**

**Comm. # 42** *L.342: I would argue that the shallow convection is mostly due to the weak updrafts and not the collision-coalescence process.*

**Response** **The sentence is revised as "The weak updraft speeds, and collision-coalescence processes mostly result in shallow convective clouds over WGs."**

**Comm. # 43** *L.480: The results now depends from rain rate as already explained. The authors should reorganized the manuscript in order to have an easy flow of useful information.*

**Response** **Thanks for the suggestion.**
**The manuscript is re-organized into different sections by synthesizing the figures. The results are discussed in detail for the figures.**

**Comm. # 44** *L.520: "remote" -> "remote sensing".*

**Response** **Referee's suggestion is implemented.**

**Comm. # 45** *L.580: "small Dm, large Nw" -> "small Dm and large Nw".*

**Response** **Referee's suggestion is implemented.**

**References:**

Das, S. K., Konwar, M., Chakravarty, K. and Deshpande, S M.: Raindrop size distribution of different cloud types over the Western Ghats using simultaneous measurements from Micro-Rain Radar and disdrometer, Atmos. Res., 186, 72–82, doi:http://dx.doi.org/10.1016/j.atmosres.2016.11.003, 2017.

Harikumar, R.: Orographic effect on tropical rain physics in the Asian monsoon region, Atmos. Sci. Lett., 17, 556-563, doi:10.1002/asl.692, 2016.

Konwar, M., Das, S. K., Deshpande, S. M., Chakravarty, K. and Goswami, B. N.: Microphysics of clouds and rain over the Western Ghat, J. Geophys. Res. Atmos., 119(10), 6140–6159, doi:10.1002/2014JD021606, 2014.

Krishna, U. V. M., Das, S. K., Deshpande, S. M., Doiphode, S. L. and Pandithurai, G.: The assessment of Global Precipitation Measurement estimates over the Indian subcontinent, Earth Sp. Sci., 4(8), 540–553, doi:10.1002/2017EA000285, 2017.

Liao, L., Meneghini, R., and Tokay, A.: Uncertainties of GPM DPR Rain Estimates Caused by DSD Parameterizations, J. Appl. Meteorol. Clim., 53, 2524–2537, doi:10.1175/JAMC-D-14-0003.1, 2014.

Radhakrishna, B., Satheesh, S. K., Narayana Rao, T., Saikranthi, K., and Sunilkumar, K.: Assessment of DSDs of GPM-DPR with ground-based disdrometer at seasonal scale over Gadanki, India, J. Geophys. Res. Atmos., 121, 11,792– 11,802, doi:10.1002/2015JD024628, 2016.

Seto, S., Iguchi, T., and Oki, T.: The basic performance of a precipitation retrieval algorithm for 5 the global precipitation measurement mission's single/dual-frequency radar measurements, IEEE T. Geosci. Remote, 51, 5239–5251, doi:10.1109/TGRS.2012.2231686, 2013.

Sumesh, R. K., Resmi, E. A., Unnikrishnan, C. K., Jash, D., Sreekanth, T. S., Resmi, M. C. M., Rajeevan, K., Nita, S. and Ramachandran, K. K.: Microphysical aspects of tropical rainfall during Bright Band events at mid and high-altitude regions over Southern Western Ghats , India, Atmos. Res., 227(March), 178–197, doi:10.1016/j.atmosres.2019.05.002, 2019.

Utsav, B., Deshpande, S. M., Das, S. K. and Pandithurai, G.: Statistical Characteristics of Convective Clouds over the Western Ghats Derived from Weather Radar Observations, J. Geophys. Res. Atmos., 122(18), 10,10-50,76, doi:10.1002/2016JD026183, 2017.

Zagrodnik, J. P., McMurdie, L. A., Houze, R. A. and Tanelli, S.: Vertical Structure and Microphysical Characteristics of Frontal Systems Passing over a Three-Dimensional Coastal Mountain Range, J. Atmos. Sci., 76(6), 1521–1546, doi:10.1175/JAS-D-18-0279.1, 2019.

---

## Author Comment (AC2) · 28 Mar 2020

**Response to Reviewer # 2's Comments**

**Overall Comment** *This manuscript presents a detailed analysis of the raindrop size distribution (DSD) during wet and dry spells of the Indian Summer Monsoon over Western Ghats of India. The DSD data are collected from a Joss-Waldvogel disdrometer during June-September of 2012-2015. DSD characteristics, including the diurnal variation and DSD spectra at different rain rates, as well as the gamma distribution parameters in different types of rainfall (i.e., stratiform vs convective) are summarized.*
*Overall, this paper is easy to follow. However, it is still not clear what the authors have addressed in addition to showing the DSD characteristics. Further discussions are required to elaborate the physical reasoning in this analysis.*

**Response** **We are indebted to the reviewer for his valuable and thoughtful comments on the manuscript. We much appreciate the reviewer's time and efforts during the evaluation of the manuscript. We went through all the referee comments and suggestions and implemented the same in the revised manuscript. Point-to-point clarifications for referee's comments and how we have addressed each recommendation is given below. The manuscript is also altered by considering the other referee's comments.**

**The present study investigates the raindrop size distribution (DSD; which provides indirect inference on rain microphysical processes such as collision, coalescence, breakup, evaporation, etc. that can shape the DSD) differences during the wet and dry spells of Indian Summer Monsoon (ISM) on the Western Ghats (WGs) region. Till now, there are limited studies of DSDs exist in the WGs region by considering long-term dataset. For the first time, the authors attempted to understand the DSD characteristics by considering the monsoon intra-seasonal oscillations (wet and dry spells). The present study brings out the results of a unique opportunity by analysing extensive data set and also considering the different phases of the monsoon intra-seasonal oscillations on the WGs region.**

**To understand the dynamical mechanisms responsible for the different DSD characteristics during wet and dry spells, the background dynamical parameters like temperature, specific humidity is studied using the ERA-Interim data along with the rain rate measurements from JWD. A separate section in the manuscript is there to discuss the dynamical processes responsible for the observed DSD differences.**

**Specific Comments**

**Comment. 1** *First of all, I do not really see anything new in this manuscript. The analysis methods are rather conventional, and the findings are not well highlighted. The authors should elaborate more on the motivation and novelty of this study. More importantly, the authors should provide sufficient discussion and explanation about the DSD characteristics indicated in the observations.*

**Response** **From the earlier studies of Ramamurthy (1969), active and break spells of**

the ISM have been extensively studied, especially during the last two decades (e.g., Goswami and Ajaya Mohan, 2001; Gadgil and Joseph, 2003; Kripalani et al., 2004; Waliser, 2006; Uma et al., 2011; Mohan and Rao, 2012; Rajeevan et al., 2013; Das et al., 2013; Rao et al., 2016). The characteristic features of these active and break spells have been well understood; for example, their identification (Rajeeven et al., 2006; Rajeevan et al., 2010), spatial distribution (Ramamurthy, 1969; Rajeevan et al., 2010; Ratan and Venugopal, 2013), circulation patterns (Goswami and Ajaya Mohan 2001; Rajeevan et al., 2010), vertical wind and thermal structure (Uma et al., 2011), rainfall variability (Deshpande and Goswami, 2014; Rao et al., 2016) and the macro- and micro-physical features of clouds (Rajeevan et al., 2013; Das et al., 2013). Even though different dynamical mechanisms for the observed rainfall distribution during the wet and dry spells are well understood, however, the microphysical processes of rain formation is still lacking.

Hence, there is a need to study the DSD characteristics using the long-term dataset, especially in the WGs region. In addition, no study aimed at understanding the DSD characteristics during monsoon intra-seasonal oscillations till now, owing to the importance of wet and dry spells on the ISM. These gaps areas motivated the authors to analyse the DSD characteristics using long-term datasets during the wetter and drier phases of the ISM over WGs. This study aims to provide a better understanding of the DSD characteristics (which provide indirect inference on rain microphysical processes such as collision, coalescence, breakup, evaporation, etc. that can shape the DSD) at these intra-seasonal time scales.

The author's used the conventional methods in the analysis because these methods are robust to understand the DSD characteristics during the wet and dry spells. Using these methods, we attempted to study the DSD characteristics for the first time during the wet and dry periods of ISM in the WGs region. This study showed a clear difference in DSDs between wet and dry spells of the ISM. These differences provide an indirect inference of rain microphysics during the wet and dry periods, also suggesting the potential requirement for different microphysical parameterization schemes in the numerical models.

The atmospheric background conditions/meteorological parameters are used to discuss the physical mechanisms responsible for different DSD characteristics during the wet and dry spells. For this, the latest reanalysis dataset, ERA-Interim, along with JWD data, is used. A separate section is included to discuss the dynamical processes responsible for the observed DSD differences over WGs.

**Comment. 2**  *In addition, I have serious concerns about using single point data to resolve the*

*orographic enhancement. Justification is required from this perspective.*

**Response**   **The authors have not discussed orographic impacts on precipitation formation and DSD using a single ground-based disdrometer. In this work, authors only discussed the DSD characteristics (indirectly rain microphysics) during the wet and dry periods of ISM. An attempt has also been made to understand the physical mechanism of DSD variations during the wetter and drier periods of ISM. Authors have used GPM products to provide a general overview of DSD variation with topography during the monsoon season. The limited dataset from GPM has restricted our further analysis of DSD variation with topography during the wet and dry spells of ISM.**

**Comment. 3**   *I am not convinced by the interpretation of the orographic gradients using GPM products. Also, the authors should incorporate the uncertainties in GPM retrievals. In fact, I do not really think this manuscript will have significant impact without including more in-situ data.*

**Response**   **This manuscript aims to examine the characteristics of DSD using disdrometer measurements during the drier and wetter periods of the ISM in the WGs region. Earlier studies have shown that the WGs precipitation processes can be modulated by the complex topography (Konwar et al., 2014; Utsav et al., 2017, etc.). Hence, in the present study, the GPM analysis is used to provide an overview of the DSD distribution over this topographic region irrespective of wet and dry spells of ISM. Such a review is not possible with single point-wise observation. Examining the orographic impact on DSD was one of the reviewer's requirements, which was addressed and included in the manuscript. The authors also believe that such analysis is useful in understanding the underlying rain microphysical processes (indirect inference on collision-coalescence, breakup, evaporation, etc. that can shape the DSD) at different altitudes. Moreover, the parameterization schemes in the numerical models also demand the dominant microphysical processes like collision-coalescence, breakup, etc. at a finer resolution. Hence, it is necessary to have the rain microphysical observations at different locations over the complex topography. At this point, the reviewer's suggestion can be considered for future work.**

**The GPM-DPR can estimate the rain integral parameters, $D_m$, and $N_w$ using dual-frequency ratio (DFR) method. The DSD parameterization used in the GPM-DPR is the gamma distribution with a constant shape parameter, μ=3. The constant value of 'μ' introduces errors in the retrievals. The retrieval of $D_m$ using the DFR method is iterative, and the $D_m$ has two solutions when the DFR is less than 0. The uncertainties in the GPM-DPR in estimating the DSD are detailed in Seto et al., (2013), Liao et al., (2014), etc. Recently, Krishna et al. (2017) assessed the DSD measurements from GPM on the WGs region by comparing them with the ground-based disdrometer. They**

showed that the seasonal variations in $D_m$ and $N_w$ are well represented in the GPM measurements, however, underestimate in comparison to the ground-based disdrometer measurement. Radhakrishna et al. (2016) also found that the GPM underestimates and overestimates the mean $D_m$ and N$w$ during the southwest and northeast monsoons over Gadanki, a semiarid region of India. In the present study, the authors intend to show the variability of DSDs over the complex terrain, not the absolute value. Hence, it is reasonable to consider the GPM measurements to present the overview of DSD over the complex topography.

The single point-wise instrument deployed at the mountain peak of WGs is not sufficient to address the orographic impacts on the DSDs. One of the difficulties in studying the effect of orography is the unavailability of the in-situ data (like disdrometer network) over the WGs region (windward and leeward slopes). It is a well-accepted suggestion to examine the effect of orography on DSDs by deploying an observational network of disdrometers and rain gauges along the windward and leeward slopes of the WGs. This suggestion can be addressed in future works.

The relative sentences have been added in the revised manuscript.

**Comment. 4** *The authors have included analyses and figures from many perspectives. But, none of them has been detailed in a very quantitative manner. In fact, the reviewer was not even clear about what problems the authors are trying to address except the detailed illustration of DSD characteristics. Significant revisions are requested to highlight the scientific merits.*

**Response** **As per the reviewer's suggestion, the authors put their sincere effort to concise the manuscript with the clear objectives and better representation of results.**

**Comment. 5** *Anyway, the authors started from an interesting point about orographic precipitation. But I did not see sufficient investigation on this aspect.*

**Response** **The motivation of the present study is to understand the DSD (which provides indirect inference on rain microphysical processes such as collision-coalescence, breakup, evaporation, etc. that can shape the DSD) characteristics at the intra-seasonal time scales over WGs. This provides an insight into the rain microphysical processes that are inferred from DSD differences between wet and dry spells precipitation. However, to have an impression of DSD over the topographic region, the authors included an overview of DSD on three different locations on the WGs. The inclusion of DSD variation with topography was also the requirement of one of the reviewers. Further non-availability of sufficient dataset, the GPM analysis is restricted to provide the overall view of the DSD irrespective of the intra-**

**seasonal oscillations. The impact of orography on the DSD distributions during wet and dry spells is important and will be taken up in future works.**

**References:**

Das, S. K., Uma, K. N., Konwar, M., Raj, P. E., Deshpande, S. M. and Kalapureddy, M. C. R.: CloudSat–CALIPSO characterizations of cloud during the active and the break periods of Indian summer monsoon, J. Atmos. Solar-Terrestrial Phys., 97, 106–114, doi:10.1016/j.jastp.2013.02.016, 2013.

Deshpande, N. R. and Goswami, B. N.: Modulation of the diurnal cycle of rainfall over India by intraseasonal variations of Indian summer monsoon. Int. J. Climatol., 34, 793–807, doi:10.1002/joc.3719, 2014.

Gadgil, S. and Joseph, P. V.: On breaks of the Indian monsoon. Indian Academy of Sciences. Earth and Planetary Sciences, 112, 529-558, 2003.

Goswami, B. N. and Mohan, R. S. A.: Intraseasonal Oscillations and Interannual Variability of the Indian Summer Monsoon, J. Clim., 14(6), 1180–1198, doi:10.1175/1520-0442(2001)014<1180:IOAIVO>2.0.CO;2, 2001.

Konwar, M., Das, S. K., Deshpande, S. M., Chakravarty, K. and Goswami, B. N.: Microphysics of clouds and rain over the Western Ghat, J. Geophys. Res. Atmos., 119(10), 6140–6159, doi:10.1002/2014JD021606, 2014.

Kripalani, R. H, Kulkarni, A, Sabade, S. S., Revadekar, J., Patwardhan, S. K., and Kulkarni, J.: Intraseasonal Oscillations during monsoon 2002 and 2003; Curr. Sci., 87, 325–351, 2004.

Krishna, U. V. M., Das, S. K., Deshpande, S. M., Doiphode, S. L. and Pandithurai, G.: The assessment of Global Precipitation Measurement estimates over the Indian subcontinent, Earth Sp. Sci., 4(8), 540–553, doi:10.1002/2017EA000285, 2017.

Liao, L., Meneghini, R., and Tokay, A.: Uncertainties of GPM DPR Rain Estimates Caused by DSD Parameterizations, J. Appl. Meteorol. Clim., 53, 2524–2537, doi:10.1175/JAMC-D-14-0003.1, 2014

Mohan, T. S. and Narayana Rao, T.: Variability of the thermal structure of the atmosphere during wet and dry spells over southeast India, Q. J. R. Meteorol. Soc., 138(668), 1839–1851, doi:10.1002/qj.1922, 2012.

Radhakrishna, B., Satheesh, S. K., Narayana Rao, T., Saikranthi, K., and Sunilkumar, K.: Assessment of DSDs of GPM-DPR with ground-based disdrometer at seasonal scale over Gadanki, India, J. Geophys. Res. Atmos., 121, 11,792– 11,802, doi:10.1002/2015JD024628, 2016.

Rajeevan, M., Bhate, J., Kale, J. D. and Lal, B.: High resolution daily gridded rainfall data for the Indian region: Analysis of break and active monsoon spells. Curr. Sci., 91, 296–306, 2006.

Rajeevan, M., Gadgil, S. and Bhate, J.: Active and break spells of the Indian summer monsoon, J. Earth Syst. Sci., 119(3), 229–247, doi:10.1007/s12040-010-0019-4, 2010.

Rajeevan, M., Rohini, P., Niranjan Kumar, K., Srinivasan, J. and Unnikrishnan, C. K.: A study of vertical cloud structure of the Indian summer monsoon using CloudSat data, Clim. Dyn., 40(3), 637–650, doi:10.1007/s00382-012-1374-4, 2013.

Ramamurthy, K.: Monsoon of India: Some aspects of the ''break'' in the Indian southwest monsoon during July and August. India Meteorological Department FMU Rep. IV-18-3, 13 pp, 1969.

Rao, T. N., Saikranthi, K., Radhakrishna, B. and Bhaskara Rao, S. V.: Differences in the Climatological Characteristics of Precipitation between Active and Break Spells of the Indian Summer Monsoon, J. Clim., 29(21), 7797–7814, doi:10.1175/JCLI-D-16-0028.1, 2016.

Ratan, R., and Venugopal, V.: Wet and dry spell characteristics of global tropical rainfall, Water Resour. Res., 49, 3830– 3841, doi:10.1002/wrcr.20275, 2013.

Seto, S., Iguchi, T., and Oki, T.: The basic performance of a precipitation retrieval algorithm for 5 the global precipitation measurement mission's single/dual-frequency radar measurements, IEEE T. Geosci. Remote, 51, 5239–5251, doi:10.1109/TGRS.2012.2231686, 2013.

Uma, K. N., Kumar, K. K., Shankar Das, S., Rao, T. N. and Satyanarayana, T. M.: On the Vertical Distribution of Mean Vertical Velocities in the Convective Regions during the Wet and Dry Spells of the Monsoon over Gadanki, Mon. Weather Rev., 140(2), 398–410, doi:10.1175/MWR-D-11-00044.1, 2011.

Utsav, B., Deshpande, S. M., Das, S. K. and Pandithurai, G.: Statistical Characteristics of Convective Clouds over the Western Ghats Derived from Weather Radar Observations, J. Geophys. Res. Atmos., 122(18), 10,10-50,76, doi:10.1002/2016JD026183, 2017.

Waliser, D.: Intraseasonal variability in Asian monsoons (ed.) Wang B, Springer-Praxis, Chichester, UK, 2006.